# Dynamic *Runx1* chromatin boundaries affect gene expression in hematopoietic development

Dominic D. G. Owens[1,4], Giorgio Anselmi [1], A. Marieke Oudelaar [1,5], Damien J. Downes [1], Alessandro Cavallo [1], Joe R. Harman [1], Ron Schwessinger [1,2], Akin Bucakci[1], Lucas Greder[1], Sara de Ornellas[1,3], Danuta Jeziorska[1], Jelena Telenius[2], Jim R. Hughes [1,2✉] & Marella F. T. R. de Bruijn [1✉]

The transcription factor RUNX1 is a critical regulator of developmental hematopoiesis and is frequently disrupted in leukemia. *Runx1* is a large, complex gene that is expressed from two alternative promoters under the spatiotemporal control of multiple hematopoietic enhancers. To dissect the dynamic regulation of *Runx1* in hematopoietic development, we analyzed its three-dimensional chromatin conformation in mouse embryonic stem cell (ESC) differentiation cultures. *Runx1* resides in a 1.1 Mb topologically associating domain (TAD) demarcated by convergent CTCF motifs. As ESCs differentiate to mesoderm, chromatin accessibility, *Runx1* enhancer-promoter (E-P) interactions, and CTCF-CTCF interactions increase in the TAD, along with initiation of *Runx1* expression from the P2 promoter. Differentiation to hematopoietic progenitor cells is associated with the formation of tissue-specific sub-TADs over *Runx1*, a shift in E-P interactions, P1 promoter demethylation, and robust expression from both *Runx1* promoters. Deletion of promoter-proximal CTCF sites at the sub-TAD boundaries has no obvious effects on E-P interactions but leads to partial loss of domain structure, mildly affects gene expression, and delays hematopoietic development. Together, our analysis of gene regulation at a large multi-promoter developmental gene reveals that dynamic sub-TAD chromatin boundaries play a role in establishing TAD structure and coordinated gene expression.

[1] MRC Molecular Hematology Unit, MRC Weatherall Institute of Molecular Medicine, Radcliffe Department of Medicine, University of Oxford, Oxford, UK. [2] MRC WIMM Centre for Computational Biology, MRC Weatherall Institute of Molecular Medicine, Radcliffe Department of Medicine, University of Oxford, Oxford, UK. [3] Physical and Theoretical Chemistry Building, Department of Chemistry, University of Oxford, Oxford, UK. [4] Present address: Structural Genomics Consortium, University of Toronto, Toronto, Ontario, Canada. [5] Present address: Max Planck Institute for Biophysical Chemistry, Göttingen, Germany. ✉email: jim.hughes@imm.ox.ac.uk; marella.debruijn@imm.ox.ac.uk

Runx1/AML1 is a member of the RUNX family of transcription factors (TFs), which are key to many developmental processes[1–3]. Runx1 is best known for its critical role in the de novo generation of the hematopoietic system and maintenance of normal hematopoietic homeostasis[1,2,4]. Disruption of RUNX1 in humans leads to several hematopoietic disorders, including acute myeloid leukemia[5] and familial platelet disorder with associated myeloid malignancy (FPD-AML)[6]. All members of the RUNX family bind the same canonical DNA motif (YGYGGT) and their tissue-specific functions are thought to be governed largely by their specific expression patterns[7]. Runx1 transcription is tightly regulated, with changes in gene dosage and expression level affecting both the spatiotemporal onset of hematopoiesis and hematopoietic homeostasis[8–11]. Runx1 is transcribed from two alternative promoters, the distal P1 and proximal P2 that are differentially regulated and generate different transcripts and protein products[12,13] (reviewed in de Bruijn and Dzierzak[1]). During hematopoietic development, Runx1 expression initiates from the P2 promoter and gradually switches to the P1 promoter with the majority of adult hematopoietic cells expressing P1-derived Runx1[12,14–17]. The Runx1 promoters do not confer tissue specificity and several distal Runx1 cis-elements have been identified that mediate reporter gene expression in transient transgenic embryos in Runx1-specific spatiotemporal patterns[16,18–21]. However, the combinatorial regulation of Runx1 at different stages of hematopoiesis is currently unclear, as are the mechanisms through which the tight and dynamic spatiotemporal control of Runx1 expression is achieved. A better understanding of Runx1 regulation may yield insights into potential avenues for therapeutic targeting of RUNX1 in a variety of hematological disorders, as recently highlighted by growth inhibition in a leukemia cell line upon loss of the Runx1+ 23 enhancer[22].

The 3D conformation of DNA in structures such as topologically associating domains (TADs) delimit the activities of enhancers in vivo[23–26]. Specific interactions achieved through chromatin folding, particularly enhancer–promoter (E-P) interactions, are thought to be a key component of spatiotemporal gene regulation[27,28]. Many insights into principles of transcriptional regulation have come from studying a few relatively small gene loci, including the globin genes. However, genes encoding developmentally important TFs, including LIM-Homeobox, Hox, Eomes, Sox, and Sonic Hedgehog (Shh), often lie in larger regulatory domains and are frequently flanked by gene deserts[29]. Indeed, studies of developmental regulators, such as Shh, have revealed exceptionally long-range enhancer–promoter interactions[30], suggesting that specific regulatory mechanisms may be at play at these large developmental loci in addition to the basic regulatory principles established at smaller genes. One aspect that remains unclear is whether large-scale chromatin conformation changes may be required to coordinate complex developmental expression patterns at larger genes. A known factor important for the regulation of chromatin conformation is CCCTC-binding factor (CTCF)[31]. CTCF, along with the loop extruding factor cohesin, mediates the establishment and maintenance of both E-P interactions and TAD structure[32–34]. Interestingly, Runx1 was mis-regulated in zebrafish after perturbation of CTCF/cohesin[35–37], suggesting that Runx1 regulation may depend on chromatin structure. Elucidating Runx1 transcriptional regulatory mechanisms is expected to contribute to a better understanding of the chromatin conformation changes employed by complex multi-promoter genes during development.

Here, we characterize the Runx1 chromatin landscape in four-dimensions, i.e., in 3D space over time, in an in vitro mouse ESC differentiation model of developmental hematopoiesis. Using high-resolution chromosome conformation capture (Tiled-C)[38], we report the presence of a pre-formed 1.1 Mb TAD spanning the Runx1 locus

in mouse ESCs that is conserved in human and forms prior to gene activation. Upon differentiation, accessible chromatin sites emerge within the TAD over known enhancers, CTCF sites, and candidate cis-regulatory elements. These regions interact with the Runx1 promoters in a developmental stage-specific manner. Notably, an increased interaction of the P1 and P2 promoters within cell-type-specific Runx1 sub-TADs is seen. These sub-TADs are bounded by highly conserved promoter-proximal CTCF sites, the role of which is poorly understood. Here, we use a machine learning approach and CRISPR/Cas9-mediated deletion to examine the importance of promoter-proximal CTCF binding for the Runx1 chromatin landscape. Deletion of either the Runx1 P1 or P2 promoter-proximal CTCF site partially disrupts TAD structure, while E-P interactions appear unaffected. Runx1 levels show a decreased trend at the mesoderm stage, concomitant with significant changes in mesodermal gene expression indicative of a delay in hematopoietic differentiation. Together, we find that sub-TAD chromatin boundaries form dynamically within the large and complex Runx1 regulatory domain during differentiation and are involved in coordinating gene expression and hematopoietic differentiation.

## Results

**Runx1 lies in a conserved TAD which forms prior to gene activation.** To investigate dynamic changes in 3D chromatin confirmation in the Runx1 locus (schematically represented in Fig. 1a) during hematopoietic development, we used the in vitro mouse ESC (mESC) differentiation model that recapitulates the de novo generation of hematopoietic progenitor cells (HPCs) from mesoderm as it occurs in the embryo, including the endothelial-to-hematopoietic transition (EHT) specific to development (adapted from ref. [39]). In this model, we assessed chromatin conformation (Tiled-C), along with gene expression (poly(A)-minus RNA-seq to capture nascent transcripts) and chromatin accessibility (ATAC-seq) over hematopoietic differentiation (Fig. 1b). Flk1+ mesodermal cells were isolated by flow cytometry from day 4 embryoid body (EB) cultures. Upon further differentiation in EHT media these gave rise to phenotypic HPCs with blood progenitor morphology and in vitro clonogenic potential (Fig. 1b, c, Supplementary Fig. 1). Gene expression analysis of mESCs, Flk1+ mesoderm, and emerging CD41+ CD45− Runx1+ HPCs reflected the developmental trajectory as visualized in a Principal Component Analysis (PCA) plot (Fig. 1d). This was accompanied by silencing of pluripotency genes (Pof5f1, Sox2, Nanog), transient expression of mesodermal genes (Flk1, Eomes, T), and increasing levels of hematopoiesis-associated genes (Pecam1, Tal1, Gfi1b, Meis1, Itga2b) (Fig. 1e). Runx1 expression was initiated in Flk1+ mesoderm and increased in HPCs (Fig. 1e[17]).

To generate high-resolution chromatin conformation maps of the Runx1 regulatory domain, we performed Tiled-C, a targeted method that generates Hi-C-like data at specific loci[38], with probes against all DpnII fragments in a 2.5 Mb region centered on Runx1. PCA of individual Tiled-C replicates showed a clear developmental trajectory (Fig. 1f, Supplementary Fig. 2) similar to that seen based on gene expression analysis (Fig. 1d), demonstrating that Runx1 exhibits reproducible dynamic chromatin conformation changes during differentiation. Runx1 resides within a 1.1 Mb TAD in mESCs (Fig. 1g, mm9 chr16:92,496,000–93,617,999) that extends to encompass the upstream 750 kb gene desert, with the Setd4 and Cbr1 genes at its telomeric end, and Clic6 at its centromeric end (Fig. 1g). Using previously published CTCF occupancy data from mESCs[40], 31 binding sites were identified within the Runx1 TAD (Fig. 1g). MEME analysis[41] identified core CTCF binding motif location and orientation underlying CTCF peaks and revealed a predominant convergence of CTCF motifs—

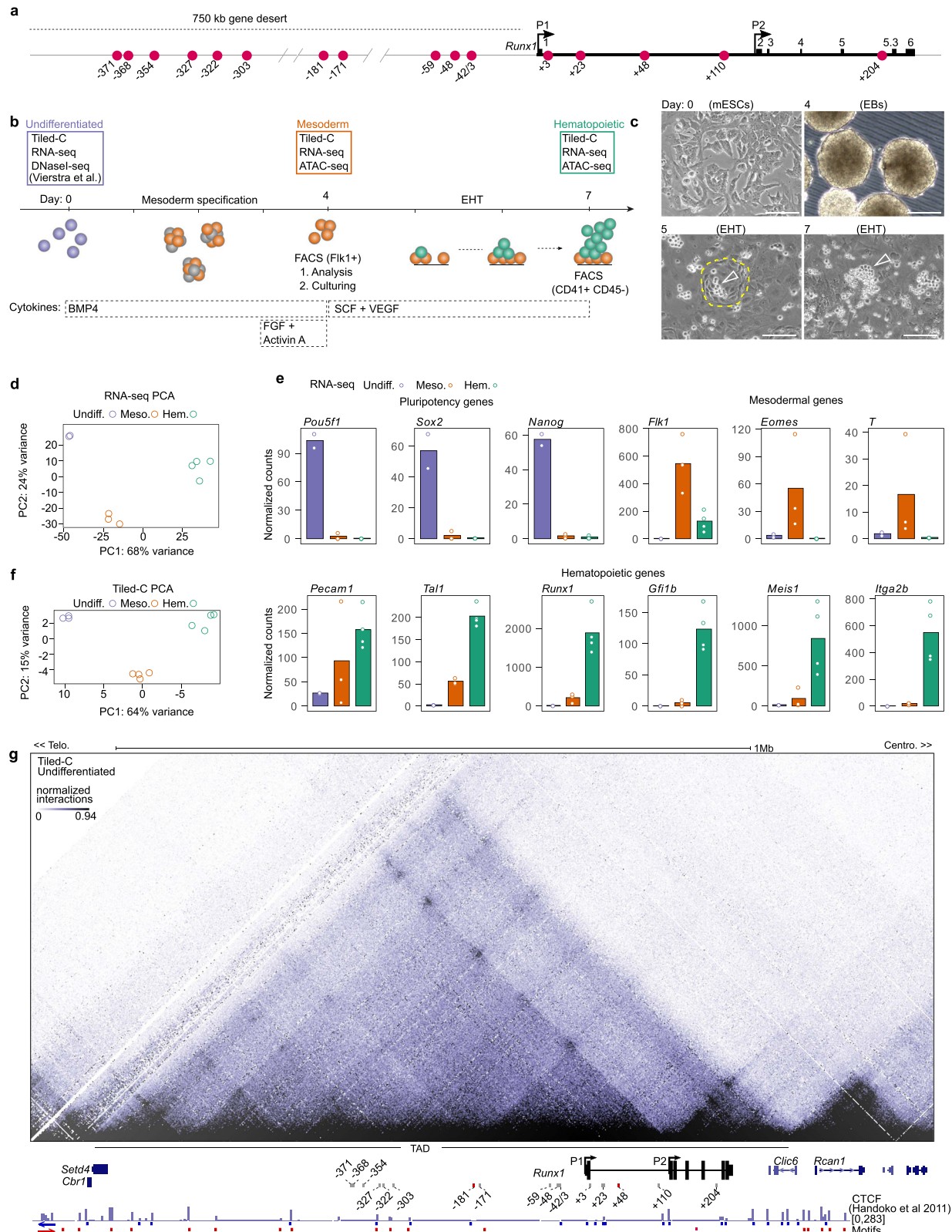

with primarily centromeric oriented motifs near the *Setd4* telomeric end of the *Runx1* TAD, and telomeric oriented motifs primarily at the *Clic6* centromeric end (Fig. 1g). Together, this shows that the *Runx1* regulatory domain is established prior to gene expression, likely in a CTCF-dependent manner.

**Mesodermal differentiation is accompanied by increased interactions between *Runx1* P2 and enhancers in the gene desert and *Runx1* gene body.** Upon transition from mESC to Flk1+ mesoderm we observed increased chromatin accessibility in the *Runx1* TAD and low levels of transcription primarily from

**Fig. 1 Runx1 resides within a topologically associating domain (TAD) in undifferentiated cells. a** Schematic of the *Runx1* locus on mouse chromosome 16, with *Runx1* proximal (P2) and distal (P1) promoters, exons, and adjacent gene desert labeled. Previously identified enhancers are indicated by red circles that are numbered according to the distance (in kb) from the *Runx1* start codon in exon 1[13,15-17,19-21,128-129]. **b** Schematic of seven-day differentiation protocol with cytokines and markers used for isolation of cells by FACS indicated. EHT = endothelial-to-hematopoietic transition. DNaseI-seq data in mESC was previously published[42]. **c** Bright-field images of different stages of in vitro differentiation. Colonies of hemogenic endothelial (HE) cells are outlined with dashed yellow lines and clusters of emerging hematopoietic progenitors are indicated by hollow white arrowheads. Scale bars = 200 μm. Representative images are shown. Experiments were performed more than ten times with similar results. **d** Principal component analysis (PCA) of individual poly(A) minus RNA-seq replicates colored by cell type. **e** Plot of normalized counts of lineage marker gene expression across differentiation. Undifferentiated $n = 2$, mesoderm $n = 3$, hematopoietic $n = 4$. **f** PCA of individual Tiled-C replicates colored by cell type. **g** Tiled-C matrix at 2 kb resolution for undifferentiated mESCs. Matrix is a merge of three independent replicates ($n = 3$). Interactions are visualized with a threshold at the 94th percentile. *Runx1* promoters (P1 and P2), neighboring genes, the adjacent gene desert, and approximate location of the 1.1 Mb *Runx1* TAD are labeled. Publicly available CTCF ChIP-seq in E14 mESCs[40] was reanalyzed and the orientation of CTCF motifs identified de novo under CTCF peaks is indicated. Previously published enhancer regions are indicated and numbered according to their distance from the *Runx1* start codon in exon 1. Enhancer regions that are accessible in undifferentiated cells are shown as red bars and enhancers that did not overlap DNaseI-seq[42] peaks are identified by gray bars.

the *Runx1* P2 promoter (Fig. 2a, b). Compared to mESCs, at this mesodermal stage we identified 36 open chromatin sites that are lost and 33 sites that are gained (DNaseI-seq from ref. [42]) (Fig. 2a, Supplementary Table 1, adjusted *p*-value <0.05). Ten accessible peaks unique to mesoderm corresponded to previously identified enhancers (−327, −322, −303, −181, −171, −59, +3, +23, +48, +110)[16,18-21,43-47] (Supplementary Table 2), while 24 peaks did not overlap with any known regulatory elements (Supplementary Table 1). Interactions between CTCF sites also increased at the mesodermal stage, particularly between the two boundaries of the *Runx1* TAD (Fig. 2c, d, Kruskal–Wallis and Dunn's test, adjusted $p = 0.003$), while insulation of the main TAD (a measure of intra-TAD interactions) decreased slightly (Fig. 2e, Kruskal–Wallis and Dunn's test, adjusted $p = 1.4 \times 10^{-135}$). To determine promoter-specific enhancer interactions, we compared virtual Capture-C profiles across the *Runx1* locus using the P2 and P1 promoters as viewpoints (Fig. 2f). We observed an overall increase of E-P2 interactions in mesoderm compared to mESCs (Fig. 2f, Kruskal–Wallis and Dunn's test, adjusted $p = 0.005$), specific increased interactions between the P2 promoter and the −327, −322, −303, −181, and −171 enhancer elements in the gene desert, and the +3, +23, and +110 enhancers in the *Runx1* gene body (Fig. 2g). In contrast to the P2, the P1 promoter did not show a significant overall increase in interactions with enhancers in mesodermal cells (Fig. 2f, Kruskal–Wallis and Dunn's test, adjusted $p = 0.9$), in line with the absence of P1-derived *Runx1* expression (Fig. 2b). However, a slight specific increase was seen in interactions between P1 and elements −181 and −171 in the gene desert, and elements +23, +48, and +110 within *Runx1* intron 1, and with the P2 could be seen (Fig. 2g). Together, these data indicate that early spatiotemporal control of *Runx1* expression at the onset of hematopoiesis is associated with increased interactions between CTCF sites, reduced TAD insulation, and may be mediated by regulation of interactions between specific enhancer elements and both the P2 and P1 promoters.

**Increased Runx1 expression upon hematopoietic differentiation is associated with P1 activation, a shift in E-P interactions, and sub-TAD reinforcement.** Differentiation of hematopoietic-fated mesoderm into HPCs is accompanied by increased expression from both *Runx1* promoters, with a three-fold higher expression from P2 than P1 (Fig. 3a, b). Compared to mesoderm, HPCs show dynamic shifts in chromatin accessibility in the *Runx1* TAD. ATAC-seq peaks were gained in the gene body at the +204 and −42 enhancers, lost at +48, −171, −181, −303, −322, and −328 and other sites in the gene desert, and maintained at the +3, +23, and +110 enhancers in HPCs (Fig. 3a, MACS2 adjusted $p < 0.05$, Supplementary Table 1). Five peaks are unique to HPCs (−42, −48.6, +6.6, +38, +204) while four peaks

are present in all three cell types (−778 [Setd4 promoter], −772 [Setd4 intronic element], −467 [gene desert CTCF site], +128 [*Runx1* P2 promoter]). Insulation of the main TAD slightly decreased further as cells differentiated from mesoderm to HPCs (Fig. 3c, Kruskal–Wallis and Dunn's test, adjusted $p = 2.1 \times 10^{-7}$), while CTCF-CTCF interactions between the boundaries of the TAD were not different from mesoderm (Fig. 3c, Kruskal–Wallis and Dunn's test, adjusted $p = 0.13$). Alongside these changes at the main TAD-level, two sub-TADs spanning the *Runx1* gene were strengthened significantly in HPCs (Fig. 3d, e, Kruskal–Wallis and Dunn's test, P1-P2 sub-TAD $p = 3.1 \times 10^{-43}$ and P2-3′ sub-TAD $p = 6.6 \times 10^{-13}$). We next compared how specific E-P interactions changed between mesoderm and HPCs using virtual Capture-C plots of the Tiled-C data. Compared to mesoderm, total E-P interactions increased in HPCs for the P2 promoter and there was a trend of increased overall E-P interactions for P1 (Fig. 3f, Kruskal–Wallis and Dunn's test, P2: adjusted $p = 0.005$, P1: adjusted $p = 0.06$). Both P1 and P2 showed specific increases in E-P interactions with −59, −43, −42, +23, and +110 enhancers (Fig. 3g), while interactions with elements extending further in the gene desert were decreased generally but maintained at non-tissue-specific CTCF sites in the gene desert. Therefore, differentiation to HPCs is associated with specific E-P interactions primarily within HPC-specific sub-TADs that span the *Runx1* gene.

**Runx1 promoter-proximal CTCF sites play a role in establishing Runx1 chromatin architecture but not E-P interactions.** The sub-TAD spanning the first intron of *Runx1* had boundaries that correspond to the P1 and P2 promoters. As both promoters have telomeric orientated CTCF sites <2 kb upstream of the transcription start site (TSS; Fig. 3a, d), and CTCF is associated with sub-TAD and TAD boundaries[31,48], we performed CTCF ChIP-seq in the 416B HPC cell line to determine if the observed changes in sub-TAD structure may be associated with differential CTCF binding in HPCs versus mESCs. Like mESC-derived HPCs, 416B HPCs express *Runx1* from both the P1 and P2 promoter (Fig. 3a, Supplementary Fig. 3a). Interestingly, while the majority of CTCF sites in the *Runx1* TAD were bound at similar levels in HPCs and mESCs, an increase in CTCF binding was seen in HPCs at the P1-proximal CTCF site, and at the +23 enhancer (Supplementary Fig. 3a). We next examined what the mechanism underlying the differential CTCF binding could be. DNA CpG dinucleotide methylation has been suggested to modulate dynamic CTCF binding[49-56], and is also known to be associated with promoter silencing[57]. To investigate whether P1 promoter methylation could underlie differential activation and CTCF binding during hematopoietic differentiation, we performed targeted bisulfite sequencing of *Runx1* promoters in undifferentiated

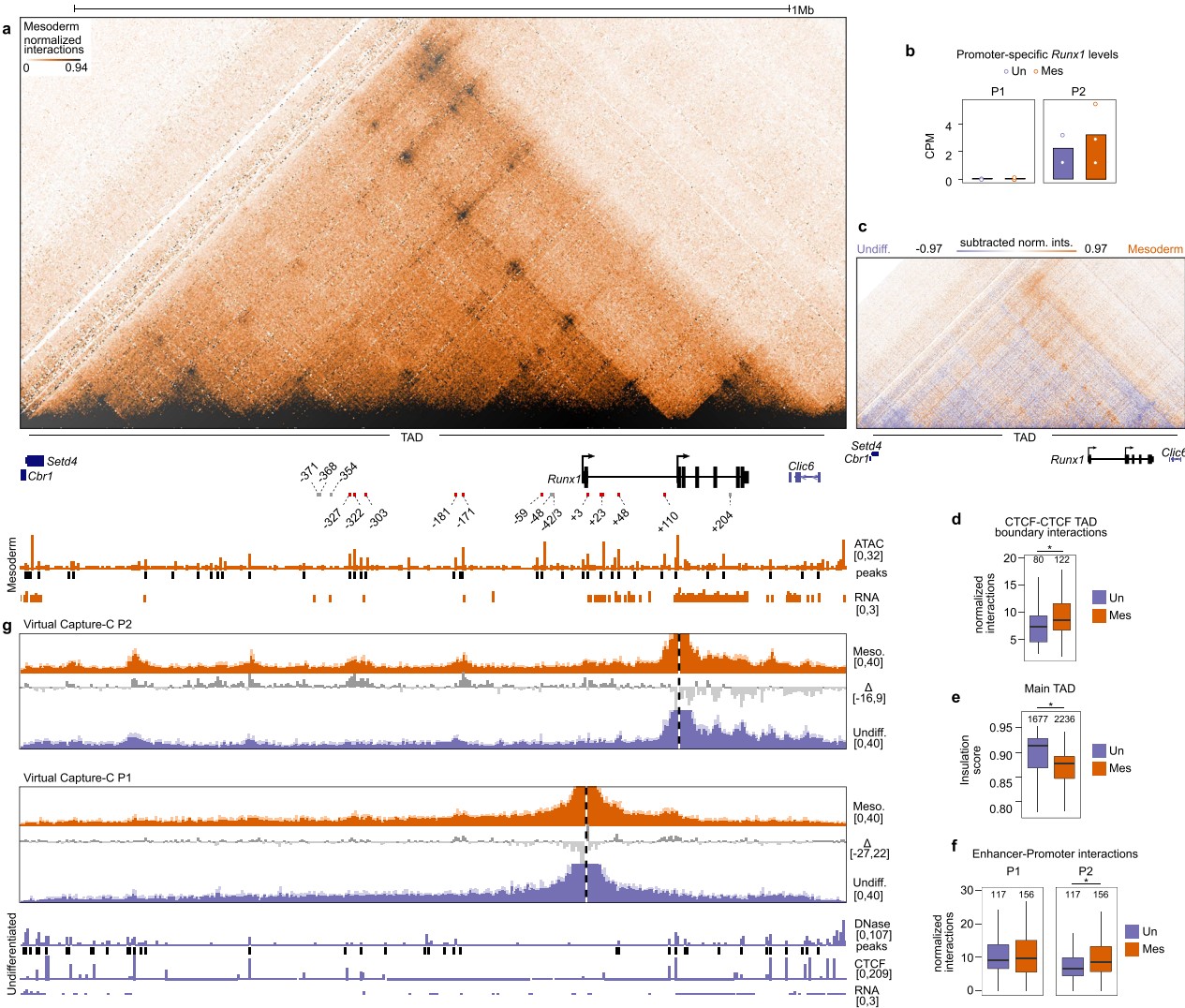

**Fig. 2 Early hematopoietic differentiation leads to increased enhancer-*Runx1* P2 interactions. a** Tiled-C matrix from mesoderm (2 kb resolution, threshold at the 94th percentile, n = 4). *Runx1* promoters and location of *Runx1* TAD are labeled below the matrix. RPKM-normalized ATAC-seq track is shown with called peaks (MACS2, adjusted p < 0.05, peaks called from one merged bam file). CPM-normalized poly(A)-minus RNA-seq (n = 3) is shown. Previously published enhancer regions are indicated. Enhancer regions that are accessible in mesoderm are shown as red bars and numbered according to their distance from the *Runx1* start codon in exon 1. Enhancers that did not overlap ATAC-seq peaks are identified by gray bars. **b** Promoter-specific *Runx1* levels in undifferentiated and mesoderm cells. Data were analyzed from two (undifferentiated) or three (mesoderm) biologically independent experiments. **c** Subtraction of normalized Tiled-C matrices between undifferentiated and mesoderm. The matrix is a subtraction of the signal between two merged matrices (undifferentiated n = 3, mesoderm n = 4, 2 kb resolution, threshold at +97th and −97th percentile). **d** Quantification of interactions between the four outermost CTCF peaks at the edges of the TAD (*, Kruskal–Wallis and Dunn's test, two-sided adjusted p = 0.003). **e** Insulation score (intra-TAD interaction ratio) of the main *Runx1* TAD (*, Kruskal–Wallis and Dunn's test, two-sided adjusted p = 1.4 × 10⁻¹³⁵). **f** Quantification of total interactions from the viewpoint of each promoter with all previously published enhancers (Supplementary Table 2) (*, Kruskal–Wallis and Dunn's test, two-sided adjusted p = 0.005). **g** Virtual Capture-C profiles (obtained from Tiled-C data, see "Methods") from the viewpoint of both *Runx1* promoters in undifferentiated mESCs (blue tracks) and mesodermal cells (orange tracks). *Runx1* promoters (P1 and P2) are indicated by a vertical dashed line. Dark colors represent the mean reporter counts in 2 kb bins (undifferentiated n = 3, mesoderm n = 4) normalized to the total *cis*-interactions in each sample. Standard deviation is shown in the lighter color. Subtractions between two cell types as indicated are shown as gray tracks. DNaseI-seq[42], DNaseI peaks, CTCF ChIP-seq[40], and RNA-seq from undifferentiated mESCs are indicated in blue below Capture-C tracks. **d–f** Boxplot centre shows median, bounds of the box indicate 25th and 75th percentiles, and maxima and minima show the largest point above or below 1.5 * interquartile range. Outlying points are not shown. Data were analyzed from the total number of bins indicated above each boxplot from three (undifferentiated) or four (mesoderm) biologically independent experiments.

mESCs and 416B HPCs. While the *Runx1* P2 promoter harbors a 2.0 kb CpG island that was hypomethylated in both cell types (Supplementary Fig. 3b), in contrast, *Runx1* P1 was near-completely methylated in mESCs, and became demethylated in hematopoietic cells (Supplementary Fig. 3b). Together, this shows that *Runx1* sub-TAD strengthening over hematopoietic

differentiation is associated with *Runx1* P1 promoter demethylation and increased CTCF binding at promoter-proximal sites.

Genome-wide, we observed a significant enrichment of CTCF binding close to active promoters in both mESCs and HPCs (<5 kb from TSS; Supplementary Fig. 3c, d, Chi-square test, p < 1 × 10⁻¹⁰). The role of promoter-proximal CTCF has not

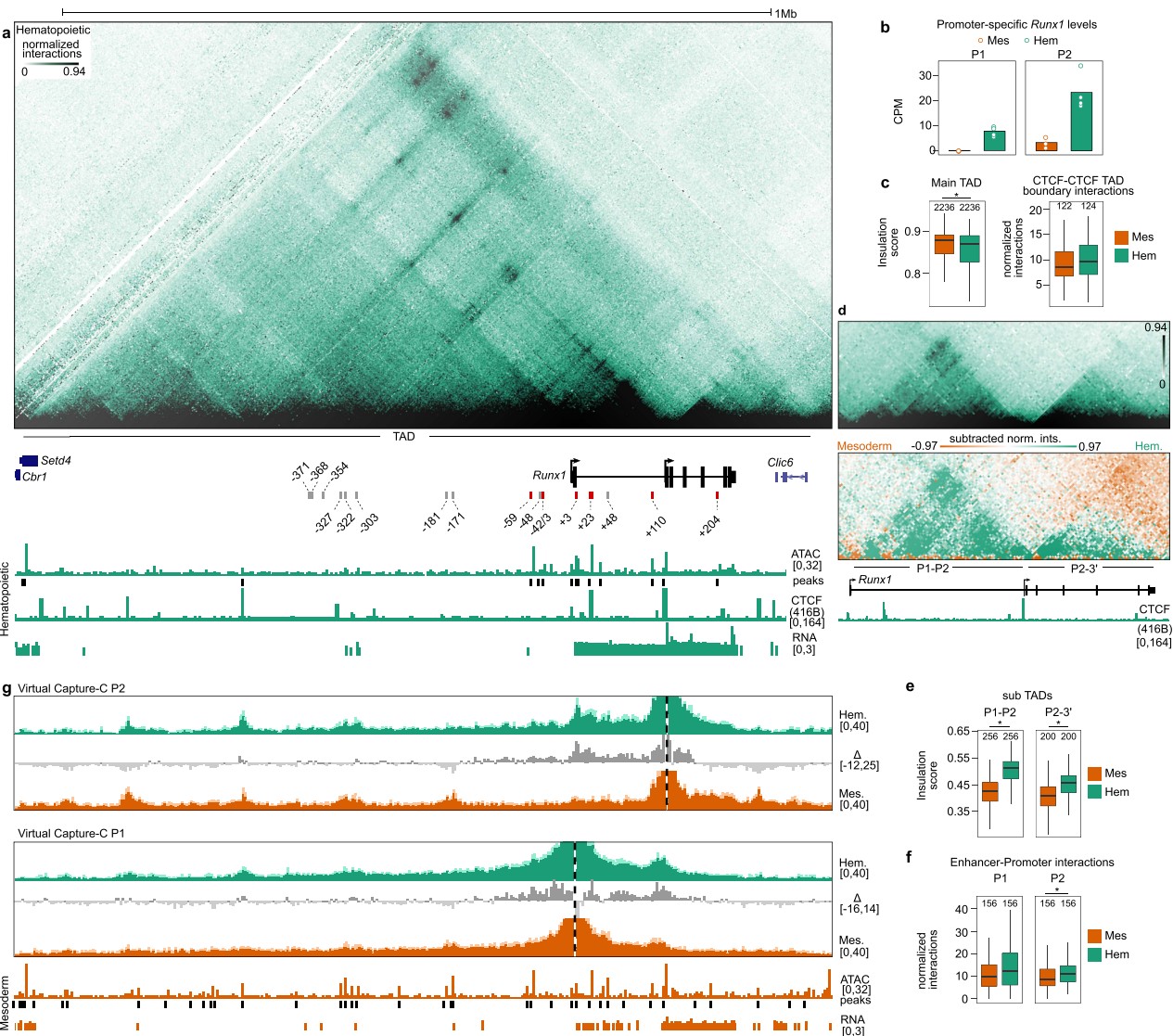

**Fig. 3 EHT progression is associated with sub-TAD reinforcement, increased *Runx1* expression and P1 activation. a** Tiled-C matrix from HPCs (2 kb resolution, threshold at the 94th percentile, *n* = 4). *Runx1* promoters and location of *Runx1* TAD are labeled below the matrix. RPKM-normalized ATAC-seq track is shown with called peaks (MACS2 adjusted *p* < 0.05, peaks called from one merged bam file). CPM-normalized poly(A)-minus RNA-seq (*n* = 4) is shown. CTCF occupancy in 416B hematopoietic progenitor cells is shown. Previously published enhancer regions are indicated. Enhancer regions that are accessible in HPCs are shown as red bars and numbered according to their distance from the *Runx1* start codon in exon 1. Enhancers that did not overlap ATAC-seq peaks are identified by gray bars. **b** Promoter-specific *Runx1* levels in mesoderm and HPCs. Data were analyzed from three biologically independent experiments. **c** Left, insulation score (intra-TAD interaction ratio) of the main *Runx1* TAD (*, Kruskal–Wallis and Dunn's test, two-sided adjusted $p = 2.1 \times 10^{-7}$). Right, quantification of interactions between the four outermost CTCF peaks at the edges of the TAD. **d** Top, zoom of Tiled-C data at 2 kb resolution with a threshold at 94th percentile. Below, subtraction of normalized Tiled-C matrices between mesoderm and HPCs. The matrix is a subtraction of the signal between two merged matrices (*n* = 4, 2 kb resolution, threshold at +97th and −97th percentile). **e** Insulation scores (intra-TAD interaction ratio) of the two *Runx1* sub-TADs (*, Kruskal–Wallis and Dunn's test, P1-P2 TAD two-sided adjusted $p = 3.1 \times 10^{-43}$ and P2-3' TAD two-sided adjusted $p = 6.6 \times 10^{-13}$). **f** Quantification of total interactions from the viewpoint of each promoter with all previously published enhancers (Supplementary Table 2) (*, Kruskal–Wallis and Dunn's test, two-sided adjusted *p* = 0.004). **g** Virtual Capture-C profiles (obtained from Tiled-C data, see "Methods") from the viewpoint of both *Runx1* promoters in mesoderm (orange tracks) and HPCs (green tracks). *Runx1* promoters (P1 and P2) are indicated by a vertical dashed line. Dark colors represent the mean reporter counts in 2 kb bins (*n* = 4) normalized to the total *cis*-interactions in each sample. Standard deviation is shown in the lighter color. Subtractions of the signal between two cell types as indicated are shown as gray tracks. ATAC-seq and peaks and RNA-seq from mesoderm are indicated in orange below Capture-C tracks. **c**, **e**, **f** Boxplot centre shows median, bounds of the box indicate 25th and 75th percentiles, and maxima and minima show the largest point above or below 1.5 * interquartile range. Outlying points are not shown. Data were analyzed from the total number of bins indicated above each boxplot from four biologically independent experiments.

been widely explored. To examine whether the deeply conserved P1 and/or P2 promoter-proximal CTCF sites (Fig. 4a), hereon referred to as P1-CTCF and P2-CTCF, may play a role in establishing the dynamic *Runx1* sub-TADs in HPCs we first utilized a deep learning approach (deepC; ref. [58]) to predict

chromatin interactions at the *Runx1* locus in mESCs. The deepC model was trained on Hi-C data from mESCs[59], withholding mouse chromosome 16 containing *Runx1*. The overall *Runx1* TAD predicted by deepC agreed well with the TAD observed in Tiled-C data from mESCs (Supplementary Fig. 4a). In silico

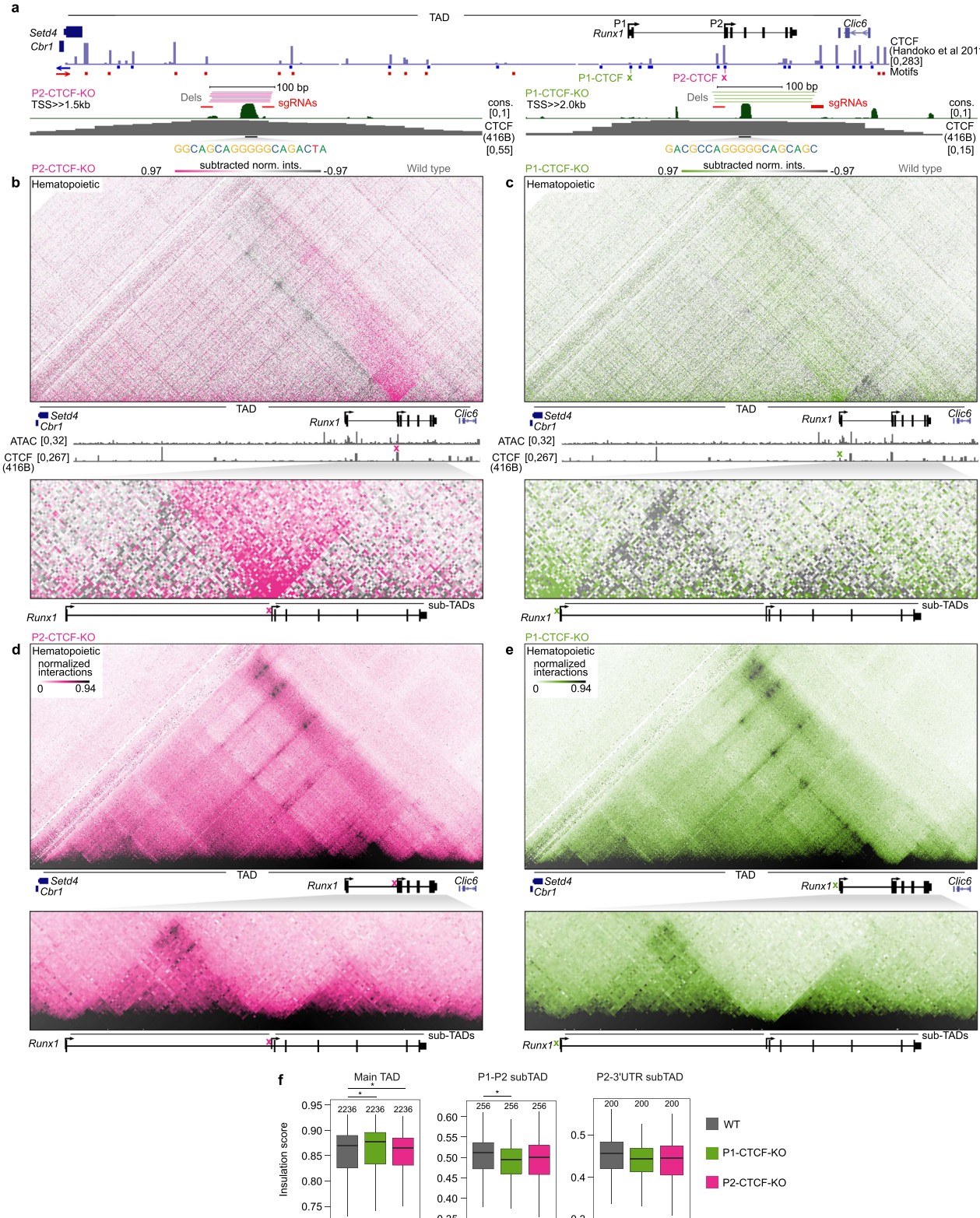

deletion of the CTCF site proximal to the P2 promoter was predicted to reduce the stripe of interactions emanating from this site into the gene desert, and to increase interactions across the boundary in mESCs (Supplementary Fig. 4b). In contrast, deletion of the P1-proximal CTCF was predicted have little effect on chromatin interactions in mESCs (Supplementary Fig. 4c).

Next, we determined the impact of promoter-proximal CTCF site deletion on chromatin conformation experimentally by Tiled-C. We

generated P1-CTCF-KO and P2-CTCF-KO mESC clones using CRISPR-Cas9; these lacked the entire CTCF site, including the core motif, but retained nearby conserved sequences (Fig. 4a, Supplementary Figs. 5–7). Hematopoietic differentiation in the KO clones was unaffected (three independent mESCs clones analyzed each for the P1-CTCF-KO and P2-CTCF-KO; Supplementary Fig. 8) and Tiled-C was performed on undifferentiated mESCs, Flk1+ mesoderm, and HPCs. PCA showed that P1-CTCF-KO and P2-CTCF-

**Fig. 4 _Runx1_ promoter-proximal CTCF sites play a role in establishing _Runx1_ chromatin architecture. a** Schematic of _Runx1_ TAD showing CTCF binding in mESCs[42] and the orientation of CTCF motifs underlying peaks. P1 and P2 promoter-proximal CTCF sites are indicated with CRISPR/Cas9 strategies to delete them. Distance to _Runx1_ transcription start sites is indicated. Vertebrate conservation (phastCons, cons), CTCF occupancy in 416B HPCs, core motif sequence, single guide (sg)RNA, and deletion alleles (dels) are indicated. **b, c** Subtraction of Tiled-C matrices between P2-CTCF-KO (**b**) and P1-CTCF-KO (**c**) and wild-type hematopoietic cells is shown at 2 kb resolution with threshold at $+/-$97th percentile of subtracted normalized interactions (subtracted norm. ints.) ($n = 4$). Locations of CTCF site deletions are indicated by a pink and green cross. RPKM-normalized ATAC-seq in wild-type HPCs and CTCF occupancy in 416B cells is shown. The locations of the main _Runx1_ TAD and sub-TADs are indicated. **d, e** Tiled-C matrix from P2-CTCF-KO (**d**) and P2-CTCF-KO (**e**) (2 kb resolution, threshold at 94th percentile, $n = 4$). **f** Insulation scores (intra-TAD interaction ratio) for main _Runx1_ TAD and sub-TADs in wild type, P1-CTCF-KO, and P2-CTCF-KO HPCs (*, Kruskal–Wallis and Dunn's test, two-sided adjusted $p$-values: main TAD WT and P1-CTCF-KO $p = 4.8^{-4}$, WT and P2-CTCF-KO $p = 0.03$, P1-P2 sub-TAD WT and P1-CTCF-KO $p = 0.003$). Boxplot centre shows median, bounds of the box indicate 25th and 75th percentiles, and maxima and minima show the largest point above or below 1.5 * interquartile range. Outlying points are not shown. Data were analyzed from the total number of bins indicated above each boxplot from four biologically independent experiments.

KO cells clustered along the same developmental trajectory as wild-type mESC differentiation cultures (Supplementary Figs. 9 and 10). Strikingly, CTCF-CTCF interactions were reduced across the entire TAD in P2-CTCF-KO HPCs, in agreement with the deepC prediction and consistent with P2-CTCF forming an insulated boundary (Fig. 4b, upper panel). Tiled-C in P1-CTCF-KO mESCs also agreed with the deepC predictions, with loss of the P1-CTCF site showing no effect compared to wild-type mESCs (Supplementary Fig. 4c). P1-CTCF-KO HPCs, however, exhibited a subtle decrease in interactions with CTCF sites in the gene desert (Fig. 4c, upper panel), highlighting the tissue-specific binding of CTCF to this site. Deletion of either P1-CTCF or P2-CTCF increased interaction frequencies between regions upstream and downstream of these sites, indicating that both CTCF sites act as boundaries (Fig. 4b and c, lower panels). Indeed, loss of P2-CTCF led to the region between _Runx1_ and _Clic6_ interacting ectopically with upstream regions (Fig. 4b, upper panel). The main TAD and tissue-specific sub-TADs were still present in P1-CTCF-KO or P2-CTCF-KO HPCs (Fig. 4d, e), though insulation scores were affected, indicating that sub-TAD boundary strengths were reduced (Fig. 4f).

In HPCs, both P1 and P2 promoters primarily interacted with enhancers lying within the tissue-specific sub-TADs (Fig. 3g). As these sub-TADs were altered upon deletion of promoter-proximal CTCF motifs, E-P interactions in P1- and P2-CTCF-KO HPCs were compared to wild-type cells. Surprisingly, despite generally weaker sub-TAD interactions in P1- and P2-CTCF-KO HPCs (Fig. 4a, b, and e), total E-P interactions were not significantly different compared to wild type for either promoter at any stage of differentiation (Supplementary Fig. 11a, b, Kruskal–Wallis test, adjusted $p > 0.4$) nor was any individual E-P interaction (Supplementary Fig. 12, Kruskal–Wallis test, adjusted $p = 1.0$). Together, our results show that despite perturbed chromatin architecture resulting from the absence of conserved promoter-proximal CTCF sites, specific _Runx1_ E-P interactions are maintained.

**_Runx1_ P2 promoter-proximal CTCF site coordinates spatio-temporal gene expression and differentiation**. Since CTCF binding close to promoters genome-wide, including at _Runx1_ P1, is associated with promoter activity (Supplementary Fig. 3c, d), and since loss of promoter-proximal CTCF sites disrupted _Runx1_ chromatin architecture, the effect of promoter-proximal CTCF loss on _Runx1_ expression was examined during hematopoietic differentiation of KO mESC clones. We observed a non-significant trend for reduced total _Runx1_ expression in P2-CTCF-KO mesoderm compared to both wild-type and P1-CTCF-KO (Fig. 5a, zoomed in graph with dashed outline, DESeq2 adjusted $p = 0.6$). No changes were observed in alternative P1 or P2 promoter usage after deletion of promoter-proximal CTCF sites (Fig. 5b). PCA of global RNA-seq profiles across all stages and genotypes showed clustering based on cell type rather than genotype (Fig. 5c). However, when considering

mesoderm samples alone, all three P2-CTCF-KO samples were located at the far end of the distribution of samples (Fig. 5d). Indeed, differential expression analysis revealed that, globally, 168 genes were differentially expressed between P2-CTCF-KO and wild-type mesoderm (Fig. 5e, DESeq2 adjusted $p < 0.05$, fold change >1). Notably, expression of several mesodermal markers (including _T_ and _Eomes_) was higher in P2-CTCF-KO mesoderm compared to wild type, while several hematopoietic markers and _Runx1_ target genes were downregulated (Fig. 5f, and Supplementary Fig. 13, adjusted $p < 0.05$)[60–62]. GO analysis of genes downregulated by P2-CTCF-KO were associated with biological processes including "response to growth factor" and "blood vessel remodeling", while upregulated genes were associated with terms including "mesoderm development" and "gastrulation" (Fig. 5g, adjusted $p < 0.05$). Collectively, this indicates that loss of the P2-proximal CTCF binding site caused a delay in in vitro hematopoietic differentiation, providing functional support for a mild decrease in P2-derived _Runx1_ transcription at or prior to the mesoderm stage.

## Discussion

In the present study, we used a cutting-edge 3C-based method to reveal dynamic changes in the _Runx1_ chromatin architecture in four-dimensions, i.e., the three-dimensional folding of chromatin over developmental time. Tiled-C[38] analysis of the _Runx1_ regulatory domain provided an unprecedented high-resolution view of _Runx1_ chromatin architecture during in vitro differentiation from mESC through Flk1[+] mesoderm to differentiating HPCs. Our detailed dissection of _Runx1_ transcriptional regulation during developmental hematopoiesis sheds light on regulatory mechanisms of complex large developmental genes. We found that _Runx1_ resides in a preformed, transcription-independent, and evolutionarily conserved main TAD that is present throughout differentiation (Fig. 6). Within this TAD dynamic sub-structures formed over development, namely sub-TADs spanning the _Runx1_ gene that appeared specifically in HPCs. We showed that promoter-proximal CTCF sites played a role in the maintenance of _Runx1_ sub-TADs, but, interestingly, not in mediating the dynamic changes in E-P interactions associated with hematopoietic differentiation. Yet, loss of the P2-proximal CTCF site led to delayed hematopoietic differentiation and disrupted gene expression specifically at the Flk1[+] mesoderm stage, possibly by slight reductions in _Runx1_ levels. Finally, we found that during hematopoietic development from mesoderm to HPCs, the _Runx1_ promoters switched from interacting with enhancers located throughout the TAD, including the gene desert, to primarily interacting with cis-elements closer to the gene and within the tissue-specific sub-TADs. This refines the region within which functional enhancers important for driving hematopoietic-specific _Runx1_ expression are likely to be found. These hematopoietic enhancers may represent therapeutic targets in leukemia, similar to what was recently shown for the _RUNX1_

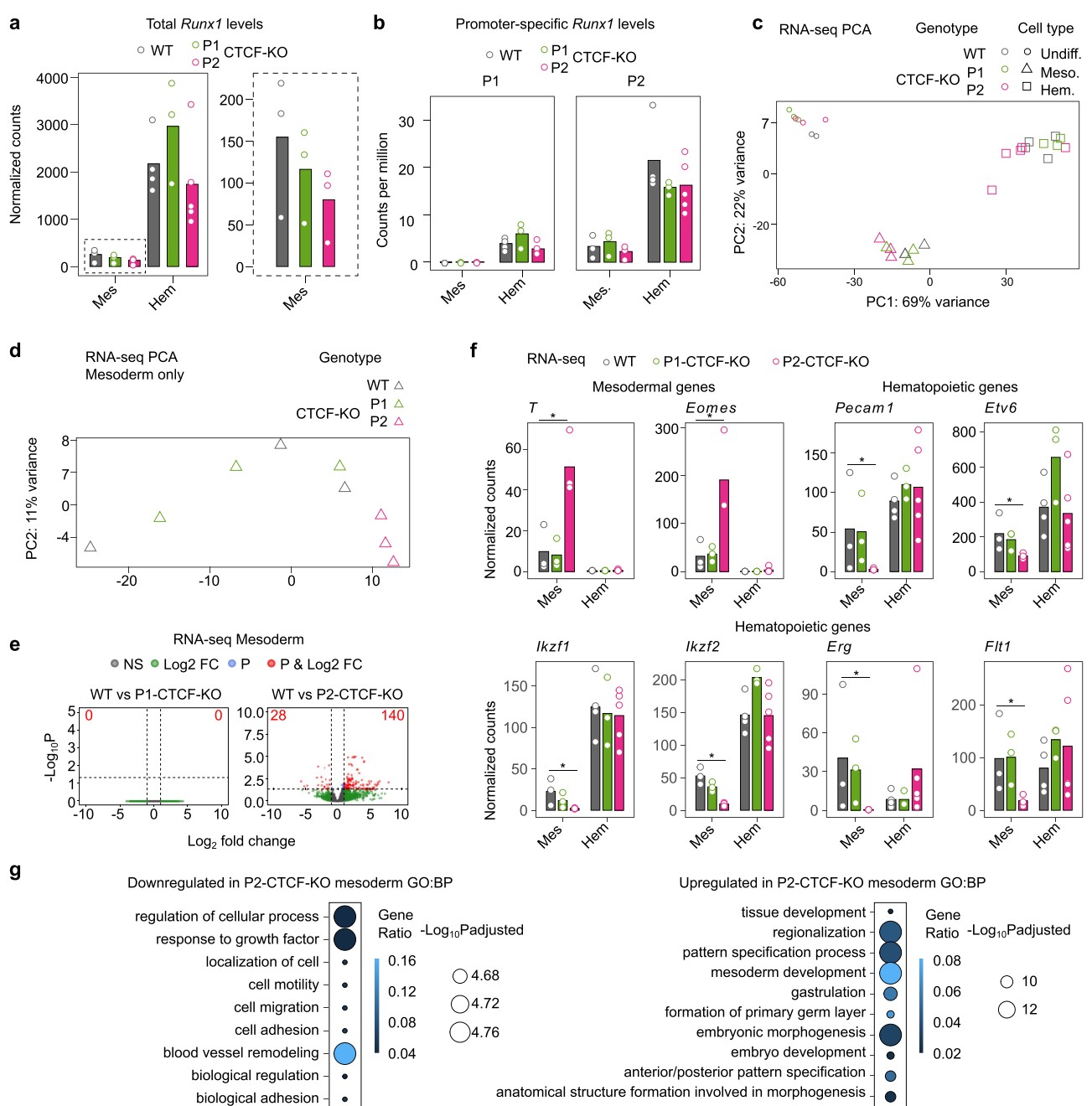

**Fig. 5 *Runx1* spatiotemporal expression is slightly altered after loss of P2-proximal CTCF. a** Total *Runx1* levels in poly(A)-minus RNA-seq in the cell types and genotypes indicated. The expanded graph with a dashed outline shows data just for mesodermal cells on a different axis. **b** Promoter-specific *Runx1* levels for each promoter in the cell types and genotypes indicated. **c** PCA of all poly(A)-minus RNA-seq replicates. **d** PCA of mesoderm RNA-seq samples. **e** Volcano plots showing differentially expressed genes (DEGs, DESeq2 adjusted two-sided $p < 0.05$, fold change >1) in P2-CTCF-KO compared to wild-type mesoderm. **f** Expression of lineage marker genes across differentiation in the genotypes indicated (*, DESeq2 adjusted two-sided *p*-values: *T* (0.048), *Eomes* (0.0021), *Pecam1* (0.037), *Etv6* (0.043), *Ikzf1* (0.016), *Ikzf2* ($1.1 \times 10^{-5}$), *Erg* (0.0079), *Flt1* (0.0078), fold change >1). **g** GO term biological processes associated with the DEG list between wild-type and P2-CTCF-KO mesoderm. Gene ratios and $-\log_{10}$ *p*-values adjusted using the Benjamini–Hochberg method are indicated for significantly enriched (goseq *p*-values adjusted with Benjamini–Hochberg procedure $p < 0.05$) GO terms. **a–g** Data were analyzed from $n = 3$ independent experiments for Wild type, P1-CTCF-KO, P2-CTCF-KO mesoderm, $n = 4$ independent experiments wild-type hematopoietic, $n = 3$ independent experiments P1-CTCF-KO hematopoietic, $n = 5$ independent experiments P2-CTCF-KO hematopoietic).

+23 enhancer[22]. As both leukemias with and without somatic or germline *RUNX1* mutations were shown to depend on wild-type RUNX1[63] (reviewed in ref. [5]) it is plausible that a similar dependency on the +23 enhancer may exist in other leukemic cells[64]. Of note, a recent report exploring GWAS studies for SNPs in *RUNX1* regulatory elements found just one mutation in the +23 enhancer and this was predicted to be pathogenic[65]. This

suggested that there is strong selection pressure to conserve the +23 enhancer in normal hematopoiesis. However, further studies are required to elucidate the dependency on *Runx1* enhancers in normal hematopoiesis and leukemia. The 4D regulatory interactions at *Runx1* described here were missed in previous reports[21,66,67] as those lacked the required high-resolution, nor included a developmental time series.

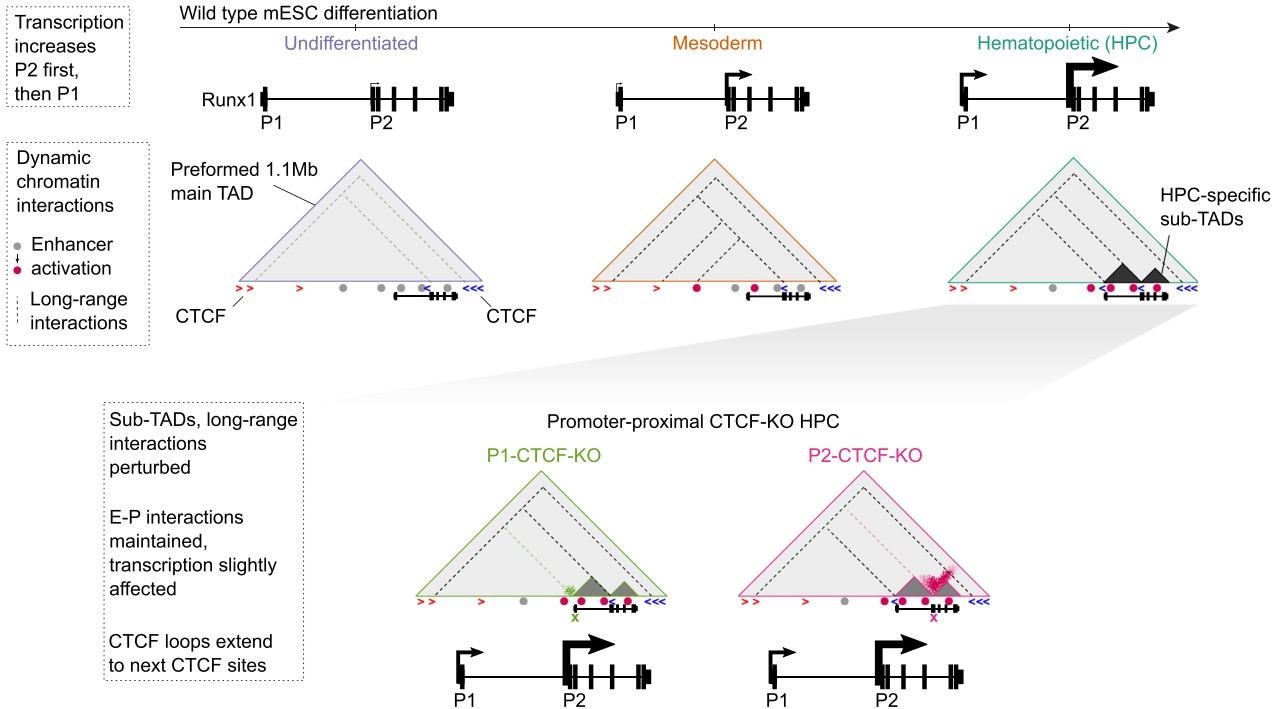

**Fig. 6 Schematic model of dynamic chromatin changes at *Runx1* during hematopoietic development and after promoter-proximal CTCF site deletions.** Large light gray triangles represent the preformed main 1.1 Mb *Runx1* TAD. The orientation of selected CTCF motifs is indicated below the TAD. Selected previously identified enhancer elements in each cell type are represented by gray circles (inaccessible) or red circles (accessible). Dashed lines throughout the TAD represent long-range chromatin interactions, with the darker color indicating a stronger interaction. Hematopoietic progenitor cell (HPC)-specific sub-TADs over the *Runx1* gene are indicated by dark gray smaller triangles within the larger TAD. A larger schematic of the *Runx1* gene is shown in each cell type with larger arrows at each promoter representing more transcription from that promoter. P1 and P2-CTCF-KO interactions in HPCs are indicated in the bottom two triangles, with the location of the deleted sites indicated by green and pink crosses. The lighter sub-TAD triangles over the *Runx1* gene indicate reduced sub-TAD insulation after promoter-proximal CTCF site deletion. The green and pink colored areas in the TAD represent increased interactions compared to wild type after deletion of P1-CTCF and P2-CTCF, respectively. The lighter green and pink dashed lines represent reduced long-range interactions from the promoters after P1 and P2-CTCF site deletion.

In line with TADs at other developmentally regulated loci[38,68–70], the overall 1.1 Mb *Runx1* TAD formed prior to differentiation and in the absence of virtually any gene transcription. The mechanism behind its establishment is likely CTCF/cohesin-mediated loop extrusion[71–74] as a predominant convergence of CTCF motifs was observed at the main *Runx1* TAD boundaries, similar to what was found at other TADs[75–77]. In addition to these preformed chromatin structures, increasing CTCF-CTCF interactions were observed upon *Runx1* activation that might reflect a higher rate or processivity of loop extrusion in the *Runx1* TAD, as was recently also observed for α-globin[38]. Alternatively, increased TF binding could be driving the specific increases in chromatin interactions that were observed over differentiation[78,79]. Two tissue-specific sub-TADs formed over differentiation, leading to a sub-compartmentalization of the *Runx1* gene itself that was correlated with promoter activation. This agrees with recent findings at alpha-globin, where gene activation is correlated with sub-TAD strengthening[38], indicating that common mechanisms underlie chromatin architecture changes during transcriptional activation at different sized gene loci. However, we also observed differences in chromatin structures between smaller and larger genes, in that sub-compartmentalization within a gene is not generally seen at smaller genes, such as α-globin, which reside entirely within one tissue-specific sub-TAD[23,80]. This difference might simply reflect the smaller size of the α-globin domain compared to the larger size of the *Runx1* regulatory domain (97 kb compared to 1.1 Mb, respectively). The sub-TAD encompassing the entire α-globin regulatory unit was shown to represent a discrete functional unit

that delimits enhancer activity[23]. We did not see evidence for this in *Runx1* as E-P interactions were not significantly changed upon sub-TAD perturbation by deleting promoter-proximal CTCF sites. However, residual sub-TAD structures may still have provided a framework within which specific regulatory interactions could take place[81]. Promoter-proximal CTCF site deletion facilitated new interactions with CTCF sites distal to the deleted CTCF sites, leading to expanded loop domains at *Runx1*. Namely, deletion of P2-CTCF led to increased interactions between upstream regions and the centromeric end of the TAD up to a cluster of CTCF sites close to *Clic6* (Fig. 6). Moreover, P1-CTCF-KO led to increased interactions upstream of the P1 promoter at the expense of downstream contacts in the P1-P2 sub-TAD which was weakened. The finding that the *Runx1* tissue-specific sub-TADs were strongest in HPCs, which express high levels of *Runx1*, suggests they may be similar to the gene-body-associated domains (GADs) recently observed at genes highly expressed in hematopoietic cells[82]. The fact that P1 and P2-CTCF sites are not in a convergent orientation, and the fact that sub-TADs/GADs form within the gene-body of actively transcribed genes, suggests that instead of being caused by a CTCF-dependent mechanism like loop extrusion, these structures may be dependent on a combination of factors including transcriptional processes[82], TF binding[78,79], and clustering of like histone modifications[83]. Indeed, our data indicate that, as reported for other promoters[58,84–86], the *Runx1* promoters may function as chromatin boundaries in a CTCF-independent manner, which would explain the residual sub-TADs and E-P interactions, as well as the expanded chromatin interactions observed in P1/P2-CTCF-KO

cells. Together, our findings suggest that loop extrusion and transcription-related mechanisms may act in concert to produce dynamic chromatin structures during differentiation.

Interestingly, deletion of the *Runx1* P2 promoter-proximal CTCF binding site resulted in a developmental delay in hematopoietic mesoderm as shown by increased expression of early mesodermal markers at the expense of later hematopoietic ones. Although studies in cell lines indicate that CTCF sites are required for TAD formation[32–34,87], interestingly, promoter-proximal CTCF sites have been suggested to play a role in E-P interactions and gene transcription[55,88–92]. Therefore, a plausible explanation for the observed differentiation delay could be that promoter-proximal CTCF loss leads to a later or perturbed onset of *Runx1* expression, as Runx1 is well known to promote hematopoietic commitment[93]. Although *Runx1* expression in the total mesodermal cell population was not significantly altered, a decreased trend was seen. This surprisingly robust *Runx1* expression and alternative promoter usage upon promoter-proximal CTCF-site loss may reflect the residual sub-TADs still found which could be attributed to redundant CTCF sites not targeted in this study (redundancy between CTCF sites has been observed before[23,58,94]), or the *Runx1* promoters themselves. An alternative explanation could be the relatively asynchronous development of cells in culture, where some cells may have had enough time to restore *Runx1* expression levels. A more substantial phenotype may be seen in a Runx1-heterozygous background, similar to what was observed in mouse embryos carrying an attenuated *Runx1* P2 allele and a non-functional *Runx1* allele, where the compound phenotype was more severe than that of a homozygous attenuated P2[16]. Finally, compared to in vitro differentiation, promoter-proximal CTCF-site loss may be more detrimental in vivo, where *Runx1* levels are subject to tight spatiotemporal control and changes to *Runx1* levels or dosage lead to knock-on effects on differentiation timing[8–11,16,95]. Together this indicates that even subtle changes in *Runx1* levels, such as the trend seen in P2-CTCF-KO mesoderm, have the potential to alter hematopoietic developmental dynamics. This underlines that *Runx1* requires an exceptionally fine-scale spatiotemporal transcriptional control and isolation from neighboring regulatory domains to support its pivotal role in development. Given that *Runx1* has important functions in development and human disease[1,2,96], an increased understanding of dynamic cis-regulatory mechanisms underpinning its regulation will be vital to future efforts to develop potential therapeutic approaches to manipulate *RUNX1* expression in human blood disorders.

## Methods

**Cell culture**. E14-TG2a mESCs[97] were cultured in GMEM medium supplemented with 100 mM non-essential amino acids, 100 mM sodium pyruvate, 10% FCS, 2 mM L-glutamine, 100 μM β-mercaptoethanol (all Gibco), and 1% Leukemia Inhibitory Factor (prepared in house). Cells were passaged using 0.05% trypsin (Gibco) every 2–3 days. The 416B mouse immortalized myeloid progenitor cell line[98] was cultured at 2–8 × 10⁵ cells/ml in Fischer's medium with 20% horse serum and 2 mM L-glutamine (all Gibco).

**Hematopoietic differentiation of mESCs**. Differentiation of mESCs was performed using a modified serum-free protocol[39,99] in StemPro-34 (SP34, Gibco) supplemented with 40X defined serum replacement, 2 mM L-glutamine (Gibco), and 0.5 mM ascorbic acid, 0.45 mM monothioglycerol (Sigma). mESCs were seeded at a density of 5 × 10⁴ cells/ml into SP34 medium plus BMP-4 (R&D, 5 ng/ml). At day 3, bFGF and Activin A were added (R&D, 5 ng/ml). At day 4, single-cell suspension was generated from embryoid bodies using 0.05% trypsin. FACS-isolated Flk1+ mesodermal cells were cultured at 5 × 10⁴ cells/cm² in SP34 plus SCF (Peprotech) and VEGF (R&D), 10 ng/ml each. After a further 3 days of culture, adherent and suspension cells were treated with 0.05% trypsin and analyzed.

**Flow cytometry and cell sorting**. Cells were stained in PBS plus 10% FCS with the antibodies listed in Supplementary Table 3. Dead cells were identified with Hoechst 33258. Cells were analyzed using a Fusion 2 flow cytometer (BD Biosciences, BD

FACSDiva Software version 8.0.1) and data analysis was performed using FlowJo (TreeStar, version 10.8).

**Colony forming unit assays**. Unsorted cells at day 6 (4 + 2) of differentiation were cultured in MethoCult 04434 (Stem Cell Technologies) in 35 mm dishes. Colonies were counted after 10 days.

**Immunocytochemistry and confocal microscopy**. Cells were plated into glass-bottom 24-well plates (ibidi) and fixed after culture for 10 min with 4% paraformaldehyde (Sigma), permeabilized, and labeled using antibodies (Supplementary Table 3) for 1 h at room temperature. Imaging was performed using a Zeiss 880 laser scanning confocal microscope.

**Deletion of CTCF sites using CRISPR/Cas9**. Single guide RNAs (sgRNAs, Supplementary Table 4) were designed (crispr.mit.edu) to flank conserved CTCF motifs. sgRNAs were cloned into pSpCas9(BB)-2A-Puro V2.0[100] containing two sgRNAs and confirmed by sequencing. mESCs were transfected with 5 μg plasmid using Lipofectamine 2000 (Invitrogen) and puromycin selected (1 μg/ml). Single colonies were isolated by limiting dilution[101]. To detect larger on-target deletions[102–104], PCR amplification was done (500 bp to 5 kb, Supplementary Table 5). Sequencing confirmed the presence of two distinct deletion alleles and the retention of DpnII restriction sites.

**Copy counting by droplet digital PCR (ddPCR)**. To further rule out the presence of undesired complex genotypes[105], copy counting was performed across the targeted regions by droplet digital PCR (ddPCR). Reactions were performed in duplex, amplifying from an internal control and a test region in every reaction. Internal control was located on mouse chromosome 4 (not targeted in these experiments and karyotypically stable in mESCs[106]). Test regions were amplified directly over the targeted region to detect loss of allele (LOA) and at 100 bp and 1 kb up- and down-stream from sgRNA target sites (Supplementary Table 5)[102]. Reactions (22 μl) contained 11 μl QX200 ddPCR EvaGreen Supermix (Bio-Rad), 25–50 ng genomic DNA purified using DNeasy Blood and Tissue Kit (Qiagen), 250 nM each of internal control primer, and 125 nM each of test primer. Standard reagents and consumables supplied by Bio-Rad were used. Ratios between test and internal control amplicons was determined in QuantaSoft software (Bio-Rad). Ratios were normalized to the mean ratio of three test amplicons located on different non-targeted chromosomes (1, 6, and 7) to determine relative copy numbers of the test amplicons.

**Chromatin interaction analysis (Tiled-C)**. Tiled-C was performed on between 7.7 × 10⁴ and 1 × 10⁶ cells using a low-input protocol[38,107,108]. Cells were cross-linked using 2% formaldehyde for 10 min. DpnII (NEB) digestion was performed shaking overnight at 37 °C. Ligation was performed overnight at 16 °C. Samples were treated with RNAse A (Roche) for 30 min at 37 °C and decross-linked using Proteinase K (Thermo Fisher) overnight at 65 °C. Digestion efficiency was quantified by qPCR (Supplementary Table 5). Libraries with >70% digestion efficiency were used (mean 83%). Up to 1 μg DNA was sonicated using a Covaris ultra-sonicator. End-repair, adapter ligation, and PCR addition of indices (7–11 cycles) was done using NEBNext Ultra II DNA library prep kit (NEB). Biotinylated capture probes 70 nt in length were designed against every DpnII restriction fragment in a 2.5 Mb window centered on *Runx1* (chr16:91,566,000–94,101,999). Probe sequences were stringently BLAT-filtered to exclude repetitive sequences, and synthesized in-house[38]. A pooled capture reaction was performed on 1 μg of each indexed 3C library. Washing of captured material was done using Nimblegen SeqCap EZ hybridisation and wash kit (Roche) and captured sequences were isolated using M-270 Streptavidin Dynabeads (Invitrogen). PCR amplification was performed for 12 cycles. Amplified DNA was purified and a second capture and PCR amplification step were performed. Libraries were sequenced on two Illumina NextSeq high-output 150 cycle runs (paired-end).

**Analysis of Tiled-C data**. Tiled Capture-C data was processed at 2 kb resolution[38]. Fastqs were analyzed using the CCSeqBasic CM5 pipeline[109] (https://github.com/Hughes-Genome-Group/CCseqBasicF/releases). Individual samples were analyzed before merging biological replicates. PCR duplicate-filtered bam files containing uniquely mapping reads were converted to sam files (samtools) and then into sparse raw contact matrices (Tiled_sam2rawmatrix.pl, https://github.com/oudelaar/TiledC). ICE normalization was done using HiC-Pro (2.11.1)[110,111] and matrices were imported into R (3.6.0). Matrices were plotted (TiledC_matrix_visualisation.py, https://github.com/oudelaar/TiledC) with a threshold between the 90th and 95th percentile. PCA was done on log normalized counts (DESeq2,1.30.1)[112]. Merged ICE normalized contact matrices were scaled to the mean number of total interactions (14,631,865) across samples using custom R scripts[113]. Virtual Capture-C plots were generated by sub-setting the matrices on individual viewpoints of interest. E-P contacts were quantified in count and ICE-normalized matrices from the viewpoint of the bin containing each promoter and bins overlapping previously published *Runx1* enhancers (Supplementary Table 2). TADs were detected by visual inspection. Intra-TAD interactions were calculated by quantifying the ratio between intra- and extra-TAD interactions for each bin within the TAD in each sample. TAD boundary contacts were quantified between

bins overlapping the four outermost CTCF sites at each of the centromeric and telomeric ends of the main TAD.

**Analysis of Hi-C data**. Publicly available Hi-C data in mESCs[59] was analyzed as previously described[38]. Data were analyzed using HiC-Pro[110] with ICE normalization[111], and plotted using python as described above.

**DeepC prediction of chromatin architecture**. Predictions of chromatin architecture were performed using deepC[58]. Briefly, deepC was trained using a transfer learning approach on distance stratified and percentile binned Hi-C data from mESCs[59], withholding mouse chromosome 16 (that contains *Runx1*) and chromosome 17 from training. Training was done in two stages. First, a convolutional neural network was trained to predict chromatin features given a 1 kb DNA sequence input. The chromatin features cover open chromatin, transcription factor binding, including CTCF, and histone modifications using publicly available DNase-seq, ATAC-seq, and ChIP-seq peaks across a range of cell types. Second, a neural network using a convolutional module followed by a dilated convolutional module was trained to predict Hi-C data given 1 Mb of DNA sequence input. The convolutional filters of the first network are used in the transfer learning to seed the filters of the convolutional module. Promoter-proximal CTCF site deletions were modeled by mutating the region spanning CRISPR/Cas9 deletion alleles that were confirmed by sequencing and predicting the chromatin interactions of the reference and deletion alleles.

**Gene expression analysis (RNA-seq)**. RNA was isolated from $1 \times 10^3$ to $2.5 \times 10^6$ cells using QIAzol (Qiagen). Total RNA was extracted using miRNeasy Mini kit (Qiagen). RNA integrity was determined using a 4200 TapeStation RNA Screen-Tape (Agilent). Ribosomal RNA was depleted from 2.5 µg total RNA per sample of undifferentiated mESC and 416B cells using the RiboMinus™ Eukaryote System v2 (Invitrogen). A poly A selection module (NEB) was used to extract poly A minus RNA and was eluted directly in First Strand Synthesis Reaction Buffer. cDNA was synthesized using the NEBNext Ultra directional library prep kit. Adapter ligation and 8–15 cycles of PCR were performed. Libraries were sequenced on Illumina NextSeq high-output 75 cycle kit (paired-end).

**RNA-seq analysis**. Fastq files were mapped to the mouse genome (mm9) using STAR (2.6.1d)[114]. PCR duplicates were removed using picard-tools (2.3.0) Mark-Duplicates. Counts per million (CPM)-normalized bigwig files were generated using deeptools (2.2.2 and 3.0.1)[115]. A blacklist file was used to exclude mapping artifacts. Bigwig files were converted to bedGraph (ucsctools (373)) and imported into R. Mean CPM was calculated for each merged sample. Reads were assigned using subread (2.0.0) featureCounts[116]. For poly(A)-minus RNA-seq data reads were assigned to both exons and introns. Assigned counts were imported into R and analyzed using DESeq2 (1.24.0)[112]. Sample clustering was performed on log normalized counts. Differential expression analysis was done using DESeq2 (adjusted two-sided p-value <0.05, fold change >1). Volcano plots were made using EnhancedVolcano (1.2.0)[117]. GO terms were calculated using goseq (1.36.0)[118] and KEGG.db (3.2.3)[119]. Gene expression was visualized using plotCounts and ggplot2 (3.3.0)[120]. Promoter-specific counts were quantified from over a 5 kb window downstream of each TSS (Runx1-P1 chr16:92,823,811–92,828,811; Runx1-P2 chr16:92,695,073–92,700,073) using bedtools (2.25.0)[121].

**Chromatin accessibility analysis (ATAC-seq and DNaseI-seq)**. ATAC-seq libraries were generated in differentiated E14-TG2a-RV mESCs (stably transfected with a Venus reporter at the 3′ end of *Runx1*[47] and a hsp68-mCherry-Runx1 +23 enhancer-reporter transgene in the *Col1a1* locus). Libraries were generated as previously described[122]. 2–5 × 10[4] differentiated cells were FACS-isolated, resuspended in cold lysis buffer, and incubated for 10 min on ice. Cells were centrifuged, supernatant discarded, and resuspended in 10 µl transposition mix. Samples were incubated for 30 min at 37 °C and quenched using 1.1 µl 500 mM EDTA. Reactions were centrifuged and incubated at 50 °C for 10 min. A total of 13 cycles of PCR were performed as in Buenrostro, Giresi, Zaba, Chang, and Greenleaf[122] with transposition reaction as a template. PCR reactions were purified using MinElute PCR purification kit (Qiagen). Libraries were sequenced using Illumina NextSeq 75 cycle kit with paired-end reads. Fastq files were mapped to the mouse genome (mm9) (NGseqBasic VS2.0)[123]. PCR duplicate-filtered bam files from individual samples (four mesoderm replicates from two experiments and two hematopoietic replicates from one experiment) were merged and filtered to remove reads mapping to chrM, ploidy regions, or the *Runx1*-Venus targeting construct (chr16:92,602,138–92,605,899, chr16:92,606,403–92,609,879), and only reads with short (<100 bp) insert sizes were retained. RPKM-normalized bigwig files were generated using deeptools (2.2.2 and 3.0.1)[115]. DNaseI-seq data in undifferentiated mESCs were downloaded from GEO (GSM1014154)[42] and analyzed as ATAC-seq data were. Peaks were called from a single merged bam file for each sample using MACS2[124] (adjusted p-value <0.05).

**CTCF binding (ChIP-seq)**. CTCF ChIP was conducted using Millipore ChIP agarose kit (Millipore). $1 \times 10^6$ cross-linked 416B cells were lysed and sonicated using a Covaris ultrasonicator. Sonicated chromatin was diluted using dilution buffer and 50 µL was removed as the 5% input control. 2 µL CTCF antibody (Supplementary Table 3) was added to 1 mL chromatin and incubated overnight at 4 °C. Decross-linking was done at 65 °C overnight. DNA was purified using phenol-chloroform-isoamylalcohol (25:24:1, Sigma) and enrichment was determined using qPCR (Supplementary Table 5). NEBNext Ultra II DNA Library Prep Kit (NEB) with 11 cycles of PCR was used to prepare sequencing libraries. CTCF ChIP libraries were sequenced using Illumina NextSeq high-output 75 cycle kit (paired-end).

**CTCF ChIP-seq analysis and de novo CTCF motif annotation**. CTCF ChIP-seq was performed in 416B cells and publicly available E14 mESC data[40] was downloaded from GEO (GSE28247). Fastq files were mapped to the mouse genome (mm9) (NGseqBasic VS2.0)[123]. De novo CTCF motifs were identified in CTCF ChIP-seq data using meme (4.9.1_1)[41,23]. CTCF peaks were called using MACS2[125] with parameters -p 0.02 using input track as a control. 2000 peaks were sampled using bedtools (2.25.0)[121] and flanking regions were extracted from the sampled peaks. Sequences of sampled peaks and flanking regions were retrieved and a background file was generated using fasta-get-markov -m 0. A de novo motif file was generated using meme with options -revcomp -dna -nmotifs 1 -w 20 -maxsize 1000000 -mod zoops. De novo motifs were identified in CTCF peaks using fimo with options -motif 1 -thresh 1e-3.

**Targeted bisulfite sequencing**. DNA methylation analysis was performed as previously described[126]. Genomic DNA (gDNA) was extracted from 1 to $5 \times 10^6$ cells using DNeasy Blood and Tissue Kit (Qiagen) and 250 ng gDNA, or Universal Methylated Mouse DNA Standard (Zymo Research) was bisulfite converted using EZ DNA Methylation-GoldTM Kit (Zymo Research). Nested PCR primer sets (Supplementary Table 5) were designed to amplify 281–379 bp overlapping target regions. External PCR reactions were performed on 1 µL bisulfite converted DNA using Hot-StarTaq DNA Polymerase (Qiagen). Internal nested PCR reactions were performed using 1 µL of the external PCR reaction. Amplicons were size selected by gel and purified. 250 ng equimolar PCR amplicons were combined for each biological sample and indexed using NEBNext Ultra II DNA Library Prep Kit for Illumina (NEB) with 6 PCR cycles. Quality of reads was assessed using fastqc/0.10.1. Reads were quality and adapter trimmed using trim galore/0.3.1 (https://github.com/FelixKrueger/TrimGalore). Reads were mapped to an in silico bisulfite converted genome using bismark/0.20.0[127]. Percentages of methylated CpG dinucleotides were determined using bismark methylation extractor. Bedgraph output files were filtered on CpG dinucleotides with coverage greater than 100 reads and imported into R. Average methylated CpG dinucleotide percentages were plotted over each region using ggplot2.

**Statistics**. All statistical tests were performed in R and were two-tailed. Tiled-C contact data were non-normal (Shapiro–Wilks test, $p < 2 \times 10^{-16}$) and so non-parametric two-sided Kruskal–Wallis test with Dunn's post hoc comparisons test was applied. Post hoc testing with Dunn's test was applied when Kruskal–Wallis test was significant and p-values were adjusted using the Holm method or the Benjamini–Hochberg method. A significance threshold of $p < 0.05$ was used for all statistical tests.

**Reporting summary**. Further information on research design is available in the Nature Research Reporting Summary linked to this article.

## Data availability
The data that support this study are available from the corresponding authors upon reasonable request. The sequencing data generated in this study have been deposited in the GEO database under accession code GSE184490. Processed Tiled-C matrices are available on github (https://github.com/d0minicO/Owens_et_al_Tiled-C). The E14 mESC CTCF ChIP-seq data used in this study are available in the GEO database under accession code GSE28247. The 416B H3K27ac ChIP-seq data used in this study are available in the GEO database under accession code GSE69776. The 416B and E14 mESC DNaseI-seq data used in this study are available in the GEO database under accession code GSE37074. Source data are provided with this paper.

## Code availability
Code is freely available on github at https://github.com/d0minicO/Owens_et_al_Tiled-C and Zenodo at https://doi.org/10.5281/zenodo.5781832[113].

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

## Acknowledgements

We thank Christina Rode, Emanuele Azzoni, Vincent Frontera, Ruth Williams, and Tatjana Sauka-Spengler for helpful discussions and advice. We thank Nick Crump and Tom Milne for expert technical assistance and advice. We thank Kevin Clark, Sally Clark, and Paul Sopp from the WIMM FACS facility for assistance with cell sorting and technical expertise. This work was supported by programs in the MRC Molecular Hematology Unit Core award to M.F.T.R.d.B. (MC_UU_12009/2) and J.R.Hu (MC_UU_00016/14), MRC MR/N00969X/1 and Wellcome Trust (106130/Z/14/Z) to J.R.Hu, Wellcome Trust Doctoral Programmes supported A.C. (108870/Z/15/Z), R.S. (203728/Z/16/Z), and A.M.O. (105281/Z/14/Z), who was also supported by the Stevenson Junior Research Fellowship (University College, Oxford), an EPSRC studentship supported J.R.Ha, a Clarendon Scholarship L.G., and S.d.O. was supported by an MRC Project Award (MR/N00969X/1) to J.R.Hu. The WIMM Flow Cytometry facility is supported by the MRC HIU, MRC MHU [MC_UU_12009], NIHR Oxford BRC and John Fell Fund [131/030 and 101/517], EPA fund [CF182 and CF170], WIMM Strategic Alliance awards [G0902418 and MC_UU_12025]. The Wolfson Imaging Centre Oxford is supported by the Medical Research Council via the WIMM Strategic Alliance (G0902418), the Molecular Hematology Unit (MC_UU_12009), the Human Immunology Unit (MC_UU_12010), the Wolfson Foundation (Grant 18272), and by an MRC/BBSRC/EPSRC grant (MR/K015777X/1) to MICA—NanoscopyOxford (NanO): Novel Super-resolution Imaging Applied to Biomedical Sciences, Micron (107457/Z/15Z).

## Author contributions

D.D.G.O., A.M.O., D.J.D., J.R.Hu. and M.F.T.R.d.B. designed and conceptualized the study. D.D.G.O., G.A., A.M.O., D.J.D., A.C., J.R.Ha. and A.B. performed experiments. D.D.G.O., A.M.O., J.R.Ha., R.S. and D.J. performed data analysis and interpretation. J.T. generated scripts used in bioinformatics analysis. L.G., S.d.O. and D.J. generated vital reagents. D.D.G.O., G.A., A.M.O., D.J.D., A.C., J.R.Hu. and M.F.T.R.d.B. wrote the paper. All authors approved of the final version to be published.

## Competing interests

J.R.Hu and D.J. are founders and shareholders of, and D.J.D. and R.S. are paid consultants for Nucleome Therapeutics. Other authors declare no competing interests.

## Additional information

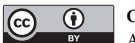

