## [Peer Review File · Nature Communications]

Dynamic Runx1 chromatin boundaries affect gene expression in hematopoietic developmentREVIEWER COMMENTS

Reviewer #1 (Remarks to the Author):

In this manuscript the authors describe the three-dimensional chromatin conformation of Runx1 during ESCs differentiating into HPCs, including chromatin accessibility, Runx1 E-P interactions and CTCF-CTCF interactions. Overall, the study is interesting and the story progresses logically, however, the authors need to address the below concerns before the manuscript is ready for publication in Nat Commun.

Major points:

1. From mESC to mesoderm, the upstream enhancers are activated in mesoderm and shows increased interaction with runx1-P2. That's great. However, why the loops still exist when the enhancers shows non-accessible in hematopoietic cells (Figure 3a). What's the function of these loops?
2. When KO p1 or p2, the E-P loops still exist (Figure 3d, e). It seems that these loops are independent of CTCF binding, which means they do not follow the loop extrusion model. What drives the formation of these loops?
3. The sub-TAD/GAD of the runx1 is interesting. What's the relationship between the sub-TADs/GADs' intensity and the expression levels of the two runx1 transcripts? Do these subTADs contribute to the usage of alternative promoters?
4. It would be better to show the differentiation potential of ESC to HPC after depletion of P1 and P2-CTCF.

Minor points:

1. The figure marks in MS are all capital, whereas in Figure panels they use lowercase. Please be consistent.
2. Please notice the usage of punctuation, such as "Supp. Figure 1" but not "Supp Figure 1".
3. If not necessary, they should delete the abbreviations, such as "cf.". What does it mean?

Reviewer #2 (Remarks to the Author):

The manuscript entitled “Dynamic Runx1 chromatin boundaries affect gene expression in hematopoietic development” by Owens et al, describes the three-dimensional chromatin conformation of the hematopoietic development essential gene Runx1 using mouse embryonic stem cells (ESC) differentiation culture. This in vitro system recapitulates the in vivo Runx1-dependent process, which occurs during embryonic differentiation leading to the emergence of hematopoietic cells from hemogenic endothelium. They defined a 1.1 Mb Runx1 topological boundary (TAD) and analyzed the role of various elements in the sub-TADs for their control of Runx1’s essential function in regulating the spatial-temporal changes in gene expression during ESC differentiation to mesoderm and then to hematopoietic progenitor cells (HPC).

This very well-written manuscript entails an excellent and very interesting study. It describes in great detail and depth the analyses of TADs by Tiled-C, including nascent RNA transcripts by poly(A)-minus RNA-seq, accessible chromatin by ATAC seq, enhancer-promoter interactions by Hi-C, H3K27ac, and CTCF Chip-seq, and CRISPR/Cas9-mediated deletion of various elements at the different stages of hematopoietic differentiation in ESC differentiation culture. No doubt that this study contributes much to our understanding of the dynamic sub-TAD chromatin structure of Runx1 and its effect on coordinating gene regulation during embryonic hematopoietic development. I am convinced that this study is of great interest to scientists in the hematopoietic development field as well as to the scientific community at large and, therefore, recommend publication in Nature Communications.

Specific comments

1. On page 3 at the end of the first paragraph, the authors mention that loss of the +23 enhancer results in growth inhibition of a leukemia cell line (Mill et al, 2019). This cell line harbors a RUNX1 mutation. It is known that a similar RUNX1 mutant leukemic cells required the remaining WT RUNX1 allele for viability. Deletion of the +23 enhancer in these leukemia cells strongly reduced RUNX1 level. Do the authors know whether conditional deletion of the +23 enhancer in adult mice also affects the growth of normal hematopoietic cells? The same question also applies to the effect of +23 deletion on the growth of other leukemic cell lines that do not carry RUNX1 mutations. These issues should be mentioned and dealt with in the Discussion.

2. On page 5 in the section Mesodermal differentiation... the authors mention the multitude of enhancers in the Runx1 TAD and their changes during differentiation. However, it is interesting to note that in addition to the 33 new ATAC sites during the transition of ESC to HE, as indicated in the MS, 36 other ATAC sites in ESC are lost in HE (Supp. Table 1). It is also worth mentioning that only 4 open

chromatin sites (ATAC sites) are unchanged in ESC, HE and HPC (-467, -772, -778 and +128) and 5 ATAC sites are unique to HPC (-42, -48.6, +6.6, +38, +204).

3. On page 6 line 5 in relation to Figure 3G, the authors mention that those specific E-P interactions (-59, -43, -42, +23, +110) were increased and that interactions with elements extending further in the gene desert were lost. However, it looks as if an interaction at approximately -400 (no coordinates shown in this region) is not lost, but rather increased. I suggest rephrasing the statement to "...interactions with some elements extending in the gene desert were lost."

4. On pages 6-7 the authors determined the effect of deleting the P1 and P2 promoter-proximal CTCF sites in ESC, on promoter-enhancer (E-P) interactions, Runx1 expression, transcriptome changes in other genes, and hematopoietic differentiation. They show that despite perturbed Runx1 chromatin architecture, resulting from the absence of conserved promoter-proximal CTCF sites, there was no change in P1 and P2 alternative usage, specific Runx1 E-P interactions were maintained, Runx1 expression level was only slightly (and not significantly) reduced and hematopoietic differentiation was unaffected. Yet, deletion of the Runx1 P2 proximal CTCF site resulted in a significant change in expression of 168 genes in the mesoderm, suggesting a delay in the in vitro differentiation of ESC to mesoderm and HPC. While the importance of Runx1 gene dosage on its biological function in hematopoiesis in vivo is a well- documented, both in Runx1/+ heterozygous mice and RUNX1 haploinsufficient humans, it is unclear how such a minor insignificant decrease in Runx1 level in Runx1 P2-proximal CTCF-KO can cause a notable gene expression changes in mesodermal cells. It is also unclear whether there is any decrease in Runx1 at the protein level in these cells. The reader will be interested to know whether deletion of Runx1 P2-proximal CTCF in ESC derived from Runx1/+ heterozygous mice (presumably causing a 50% reduction in Runx1 level) will affect mesodermal gene expression during their in vitro differentiation to the same extent or higher, compared to ESC derived from Runx1 WT mice. It would also be of interest to find out whether Runx1 P2-proximal CTCF-KO mice will display hematopoietic defects in the megakaryocytic lineage as occur in Runx1/+ mice. These points should be addressed and discussed.

Reviewer #3 (Remarks to the Author):

Runx1 is a transcription factor that plays a crucial role in definitive haematopoiesis. Genetic events that affect the RUNX1 gene in humans contribute to the disease pathology of myeloid leukaemias. Therefore, control of Runx1 expression is of high interest to clinicians and biomedical scientists researching blood development and leukaemia. In this interesting paper, Owens et al investigate the

contribution of 3D genome organisation to Runx1 expression in mouse ES cells, during a process in which these cells are induced to differentiate into mesoderm and then into haematopoietic progenitor cells (HPC). They found that connections from the two Runx1 promoters alter as the cells differentiate, but that deletion of sites that recruit CTCF at each of the two promoters of Runx1 make little difference to enhancer-promoter connections, with the main effect being a delay in haematopoietic differentiation. This is a beautifully conducted study that was a pleasure to read, and my comments are limited to data interpretation and presentation.

Major comments

When illustrating how the connections change between the Runx1 promoters and the enhancers, it would be good to have consistency in the schematics underneath the TAD diagrams in the figures (2-3, under the 'a' panel of each). Each figure should represent the same set of enhancers, rather than a subset. The enhancers shown should exactly reflect those shown in Fig 1a.

Because the P1 and P2 CTCF motifs are in the same (reverse) orientation, you wouldn't expect CTCF-anchored loops to be responsible for the interactions between P1 and P2 (same-oriented CTCF sites don't stop loop extrusion), ie the sub-TADs aren't likely to be CTCF-dependent anyway. The contacts that show up on the tiling Hi-C could be being mediated by TFs or compartment formation (clustering of like histone modifications).

KO of CTCF sites. It's actually not surprising that taking out CTCF sites at P1 and P2 didn't affect expression or E-P contacts very much, because the contacts during differentiation are likely to be driven by TFs. The data on these CTCF site KOs could be interpreted more specifically. For example - deletion of the P2 CTCF site allows from P2 onwards to the 3' end of the gene to be incorporated into larger looped domains that now terminate near *Clic6*. That is, CTCF would likely bind to the next best 3' CTCF site. In both cases this should expand the loops size that P1 and P2 can reside in, and influence the frequency of local contacts. Deletion of the P1 CTCF site allows P1 to make contacts to the 5' that seem to come at the expense of 3' contacts down to P2 (because the P1-P2 sub-TAD is weakened - P1 seems to fall out of this region).

The authors should take the opportunity include a model diagram/figure, illustrating pictorially what changes happen in the 3D organisation of the Runx1 gene as cells differentiate. Diagrams could be included to show the expansion of TADs/loops downstream to near *Clic6* upon deletion of the P2 CTCF site. Expansion of these loops into larger loops could weaken E-P interactions that control transcription bursts at P2, thereby leading to the delay in haematopoietic development that the authors observed.

Specific comments

Tiling Hi-C - works by having a probe to every unique DpnII fragment. There are some repetitive regions in the mouse Runx1 TAD. Can the frequency of interactions at repetitive regions be confidently mapped?

Figure 1 d-f - there seem to be different numbers of replicates for undifferentiated, mesoderm, and HPC samples - explanation for this?

Fig 1e, Flk1 is the term used for this gene in the article text while Kdr is used in the figure, please make consistent between the two.

Fig 1g, the lack of annotated 'forward' CTCF motifs under P2 suggests that the loop/domain between P2 and the Clic6 boundary is not CTCF-dependent, at least not at the 5' end. Do the authors have alternative explanations?

Fig 2a, enhancers and ATAC peaks, the enhancer annotations are not all the same as those listed in the red circles in fig 1a. There seems to be extra ones in fig 2a. In fact, it would be good to keep all elements consistent between figs 1-3 in the diagrams beneath the TAD images.

Fig 2g P1 appears to also interact more with +23 upon differentiation to mesoderm.

Page 5: "early spatiotemporal control of Runx1 expression at the onset of hematopoiesis is associated with increased CTCF interactions" - the authors haven't determined that these interactions are mediated by CTCF. They could be TF-mediated or compartment-mediated.

Fig 3a, it would be helpful to label in the diagram under the TAD image, the more distal enhancers, -371, -368 etc. It looks like P2 hangs onto one of these when differentiating to HPCs.

Page 7: "highlighting the tissue-specific nature of this CTCF site." should rather refer to the differential binding of CTCF at this site, because the DNA motif remains unchanged.

Fig 4a, It would be good to reproduce the schematic from Fig 1g (given below the TAD diagram) showing specifically which of these CTCF sites was deleted.

Fig 4b, P2-CTCF KO, the enrichment of 'pink' pixels downstream of this deleted CTCF site indicates expansion of interactions such that the Runx1 gene down to near Clic6 is now contained within several of the upstream anchored loops.

Fig 4c, P1-CTCF KO, there's a loss of interactions in the P1-P2 intron and a gain of interactions proximal to P1 at the 5' end of the site - this could be due to loss of shielding of P1 from nearby 5' enhancers.

Fig 5d, stretching it a bit to say that the CTCF-P2 KO is clustering dramatically differently, because the other conditions don't cluster or separate out that much.

Fig 5f, surprising that pluripotency genes are up in both P1 and P2 CTCF-KO lines in undifferentiated cells, why is this?

Page 8: " In addition to these preformed chromatin structures, increasing CTCF-CTCF interactions were observed upon Runx1 activation that might reflect a higher rate or processivity of loop extrusion in the Runx1 TAD." Very unlikely to be the mechanism, more likely to be TF-driven, for a recent confirmatory ref see PMID: 32213323. Consider the view that TADs merely provide a framework for specific regulatory interactions, like E-P interactions - see PMC8035076

We thank the reviewers for their constructive and insightful comments, which we believe have helped to improve our manuscript.

Reviewer #1 (Remarks to the Author):

In this manuscript the authors describe the three-dimensional chromatin conformation of *Runx1* during ESCs differentiating into HPCs, including chromatin accessibility, *Runx1* E-P interactions and CTCF-CTCF interactions. Overall, the study is interesting and the story progresses logically, however, the authors need to address the below concerns before the manuscript is ready for publication in *Nat Commun*.

Major points:

1. From mESC to mesoderm, the upstream enhancers are activated in mesoderm and shows increased interaction with *runx1*-P2. That's great. However, why the loops still exist when the enhancers shows non-accessible in hematopoietic cells (Figure 3a). What's the function of these loops?

It is clear from our data at *Runx1*, and from previous publications (e.g. Brown et al. 2018, Hug et al. 2017, Oudelaar et al. 2020, Paliou et al. 2019) that TAD structures and chromatin interactions exist independently of transcription. These interactions in the gene desert are likely driven by non tissue-specific CTCF sites that maintain interactions with the promoters despite decreased gene desert enhancer accessibility. This interesting point is now addressed directly in the text (page 6, line 9-10), where we indicate that “interactions with elements extending further in the gene desert were **decreased generally but maintained at non-tissue-specific CTCF sites in the gene desert**” In conclusion, we believe that these interactions are structural in nature and are unlikely to be directly related to gene expression.

2. When KO p1 or p2, the E-P loops still exist (Figure 3d, e). It seems that these loops are independent of CTCF binding, which means they do not follow the loop extrusion model. What drives the formation of these loops?

This insightful comment provides us an excellent opportunity to clarify our thinking on the role of loop extrusion in the chromatin interactions observed in our Tiled-C data. The loop extrusion model proposes that a loop extruding factor (likely cohesin) translocates along chromatin until a boundary element halts this translocation. Typically, chromatin boundaries are thought to be located at convergently oriented CTCF sites. However, ever since TADs were first observed in Hi-C data, chromatin boundaries have also been observed at actively transcribed regions such as promoters (Dixon et al. 2012; PMID: 22495300; Rao et al. 2014; 25497547). Further, recent work that was published after we submitted our manuscript demonstrates that active promoters can act as orientation-dependent chromatin boundaries (Bozhilov et al. 2021; PMID: 34155213). Together, this supports our finding that *Runx1* E-P interactions are maintained after promoter-proximal CTCF loss, as loop extrusion functions to bring E-Ps together independent of promoter-proximal CTCF binding. We added this new citation to the discussion (page 9, line 28-31) and clarified that we found that “as reported for other promoters (Bozhilov et al. 2021, Cho et al. 2018, Harrold et al. 2020, Schwessinger et al. 2020), the *Runx1* promoters may function as chromatin boundaries in a CTCF-independent manner, which would explain the residual sub-TADs **and E-P interactions, as well as the expanded chromatin interactions** observed in P1/P2-CTCF-KO cells”

3. The sub-TAD/GAD of the *runx1* is interesting. What's the relationship between the sub-TADs/GADs’

intensity and the expression levels of the two *runx1* transcripts? Do these subTADs contribute to the usage of alternative promoters?

Firstly, we agree with the reviewer that the sub-TADs observed over *Runx1* represent an interesting finding worth highlighting. Sub-TADs strengthen as differentiation progressed from mesoderm to HPCs (Figure 3e and in the text, page 6, line 1-3) and this strengthening was associated with increased expression from both *Runx1* promoters (Figure 3b and the final paragraph on page 5). This agrees with recent findings at alpha-globin, where gene activation is correlated with sub-TAD strengthening (Oudelaar, Beagrie et al. 2020; PMID: 32483172). We conclude that *Runx1* sub-TADs are correlated with promoter activity during gene activation and have added a sentence to this effect in the discussion (page 9, line 1-3), which reads “Two tissue-specific sub-TADs formed over differentiation, leading to a sub-compartmentalization of the *Runx1* gene itself that was correlated with promoter activation. This agrees with recent findings at alpha-globin, where gene activation is correlated with sub-TAD strengthening”.

Secondly, we can conclude that disruption of sub-TADs by promoter CTCF deletion did not change *Runx1* alternative promoter usage (page 7, line 38-40) as “No changes were observed in alternative P1 or P2 promoter usage after deletion of promoter-proximal CTCF sites (Figure 5b)”. Therefore, alternative promoter usage appears not to be regulated by promoter-proximal CTCF sites, but may still be regulated by residual sub-TADs formed by CTCF-independent mechanisms. The discussion has been modified to highlight this interpretation of the data (page 9, line 44-45), now reading “robust *Runx1* expression and alternative promoter usage upon promoter-proximal CTCF-site loss may reflect the residual sub-TADs”

4. It would be better to show the differentiation potential of ESC to HPC after depletion of P1 and P2-CTCF.

While there is general agreement in the field that there can be redundancy between CTCF sites (addressed in, for example, Hanssen et al. 2017; PMID: 28737770), we decided against deleting both P1 and P2 CTCF sites for the following reasons:

1) The focus of this study is on the dynamic 3D chromatin changes seen at *Runx1* during hematopoietic differentiation, including changes in E-P interactions. Based on the intriguing observation of sub-TAD formation during differentiation, we extended our study to explore a possible role of promoter proximal CTCF sites in this, as promoter proximal CTCF sites are a previously not well studied class of CTCF sites. While in theory CTCF site redundancy might explain the mild phenotype observed with P1 or P2 CTCF site deletion (discussed on page 9, line 44-48), the P1 and P2 CTCF sites are in the same orientation making it unlikely that they are together responsible for the sub-TAD formation, as also commented by reviewer 3. This information has been added to the discussion on page 9 (line 23-27) to read “The fact that P1 and P2-CTCF sites are not in a convergent orientation, and the fact that sub-TADs/GADs form within the gene-body of actively transcribed genes, suggests that instead of being caused by a CTCF-dependent mechanism like loop extrusion, these structures may be dependent on a combination of factors including transcriptional processes (Zhang et al. 2020), TF binding (Hsieh et al. 2020, Hua et al. 2021), and clustering of like histone modifications (Ruthenburg et al. 2007).”

2) To properly address the question of CTCF site redundancy, a systematic perturbation of CTCF sites would need to be undertaken. There are thirty-one CTCF sites in the *Runx1* TAD, several of which could play a role in redundantly regulating *Runx1*. Perturbing these would be a large undertaking, as for each (combination of) CTCF site(s), at least three independent clones would have to be generated and fully

validated on multiple levels (as was done in Supp. Figures 5 and 6) to ensure the integrity of the locus. We feel this would be a separate project, outside the scope of our study.

Minor points:

1. The figure marks in MS are all capital, whereas in Figure panels they use lowercase. Please be consistent.

All figure panels are now referred to in the text with lowercase letters.

2. Please notice the usage of punctuation, such as “Supp. Figure 1” but not “Supp Figure 1”.

The typographical error on page 4 (line 34) has been corrected and now reads “Supp. Figure 1”

3. If not necessary, they should delete the abbreviations, such as “cf.”. What does it mean?

We have deleted the abbreviation “cf” on page 4 (line 45) and page 7 (line 22).

‘cf.’ stands for the Latin ‘confer/conferatur’, which means ‘compare’ and is used in writing to refer the reader to other material to make a comparison with the topic being discussed.

Reviewer #2 (Remarks to the Author):

The manuscript entitled “Dynamic Runx1 chromatin boundaries affect gene expression in hematopoietic development” by Owens et al, describes the three-dimensional chromatin conformation of the hematopoietic development essential gene Runx1 using mouse embryonic stem cells (ESC) differentiation culture. This in vitro system recapitulates the in vivo Runx1-dependent process, which occurs during embryonic differentiation leading to the emergence of hematopoietic cells from hemogenic endothelium. They defined a 1.1 Mb Runx1 topological boundary (TAD) and analyzed the role of various elements in the sub-TADs for their control of Runx1’s essential function in regulating the spatial-temporal changes in gene expression during ESC differentiation to mesoderm and then to hematopoietic progenitor cells (HPC).

This very well-written manuscript entails an excellent and very interesting study. It describes in great detail and depth the analyses of TADs by Tiled-C, including nascent RNA transcripts by poly(A)-minus RNA-seq, accessible chromatin by ATAC seq, enhancer-promoter interactions by Hi-C, H3K27ac, and CTCF Chip-seq, and CRISPR/Cas9-mediated deletion of various elements at the different stages of hematopoietic differentiation in ESC differentiation culture. No doubt that this study contributes much to our understanding of the dynamic sub-TAD chromatin structure of Runx1 and its effect on coordinating gene regulation during embryonic hematopoietic development. I am convinced that this study is of great interest to scientists in the hematopoietic development field as well as to the scientific community at large and, therefore, recommend publication in Nature Communications.

Specific comments

1. On page 3 at the end of the first paragraph, the authors mention that loss of the +23 enhancer results in growth inhibition of a leukemia cell line (Mill et al, 2019). This cell line harbors a RUNX1 mutation. It is known that a similar RUNX1 mutant leukemic cells required the remaining WT RUNX1 allele for viability. Deletion of the +23 enhancer in these leukemia cells strongly reduced RUNX1 level.

- Do the authors know whether conditional deletion of the +23 enhancer in adult mice also affects the growth of normal hematopoietic cells?
- The same question also applies to the effect of +23 deletion on the growth of other leukemic cell lines that do not carry RUNX1 mutations.

These issues should be mentioned and dealt with in the Discussion.

The reviewer raises interesting questions. Regarding the role of the +23 enhancer in adult mouse hematopoiesis, we unfortunately do not have data to directly answer this as we have not generated enhancer deleted mouse models. However, there is circumstantial evidence to suggest that the +23 enhancer may be required for correct spatiotemporal and/or levels of expression also in adult hematopoiesis. First, the +23 enhancer-GFP reporter mouse we generated does label virtually all adult BM cells, apart from the maturing erythroid lineage, similar to what is seen in Runx1-reporter KI mouse models. Second, the *Runx1* +23 enhancer is accessible in almost all cell types of the mouse adult hematopoietic/immune system including LT-HSCs, lymphoid, and myeloid lineages (Rebuttal Figure 1, generated from data from Yoshida et al. 2019; PMID 30686579). Third, a recent paper by the Horsfield

group (Thomas et al., 2021; PMID 34440349), published after our manuscript was submitted, explored GWAS studies for SNPs in *RUNX1* regulatory elements. While they found multiple SNPs associated with other *RUNX1* enhancers, none with a minor allele frequency >1% were found in the +23 enhancer. There was just one reported mutation for the +23 enhancer (chr21:g.36399191A>T) that was predicted by FATHMM (Functional Analysis through Hidden Markov Models) to have a pathogenic function. The authors concluded that this suggests that there is strong selection pressure to conserve the +23 enhancer. Altogether, these findings suggest that the +23 enhancer plays a role in regulating *Runx1* expression in adult hematopoiesis, but to the best of our knowledge this remains to be experimentally verified.

Rebuttal Figure 1. ATAC-seq tracks over the mouse *Runx1* locus in hematopoietic cells (data from Yoshida et al., 2019; PMID 30686579).

ATAC-seq enrichment tracks are shown over the *Runx1* locus derived from various adult mouse hematopoietic lineages. The two *Runx1* promoters (P1 and P2) and the +23 hematopoietic enhancer are indicated. Bone marrow-derived hematopoietic stem and progenitor cell (HSPC) types indicated are long-term repopulating hematopoietic stem cells (LT-HSC) and multipotent progenitors (MMP). Lymphoid lineage cells indicated are splenic B cells (B) and natural killer T cells (NKT), and T cells derived from thymus (CD4 and CD8 single positive). Myeloid lineages indicated are granulocyte (Gran.), macrophage (Mac) derived from spleen, and monocytes (Mono.) derived from peripheral blood. For sort markers please refer to Yoshida et al., 2019.

Regarding the second point, it has indeed been shown that leukemic cells not carrying a *RUNX1* mutation also require wild type *RUNX1* for their survival (Ben-Ami et al., 2013, PMID: 24055056;

reviewed in Sood et al., 2017, PMID: 29348313), which together with the study by Mills et al., 2019 raises the question of the effect of +23 deletion on the growth of other leukemic cell lines that do not carry RUNX1 mutations. Such an extended requirement for the +23 enhancer in other leukemias is plausible, given a recent study perturbing (rather than deleting) enhancer activity by treatment with the BET inhibitor JQ1. Anthony et al., 2020 (PMID 31897485) suggest that “cohesin-STAG2 depletion [in K562 cells that don’t carry a RUNX1 mutation] de-constrains the chromatin surrounding RUNX1 and ERG, which causes aberrant enhancer-amplified transcription in response to differentiation signals.” And “showed that enhancer suppression using BET inhibitor JQ1 prevents aberrant RUNX1 and ERG signal-induced transcription in STAG2 mutant cells and reduces leukaemic stem cell characteristics of STAG2 mutants”. Notably, the +23 enhancer is bound by Brd4 in K562 cells and binding is reduced after JQ1 treatment (Lie et al., 2017; PMID: 28841410), suggestive of a role for the +23 also in regulating RUNX1 expression in leukemic cells not carrying a RUNX1 mutation. Finally, Buijs et al., 2012 (PMID 22430633) suggested that loss of the +23 enhancer in a translocation associated with FPD/AML not harbouring RUNX1 coding mutations may have contributed to the phenotype.

We have commented on these points in the Discussion (page 8, line 29-36): “As both leukemias with and without somatic or germline RUNX1 mutations were shown to depend on wild type RUNX1 (Ben-Ami et al., 2013; reviewed in Sood et al., 2017) it is plausible that a similar dependency on the +23 enhancer may exist in other leukemic cells (Anthony et al., 2020). Of note, a recent report exploring GWAS studies for SNPs in RUNX1 regulatory elements found just one mutation in the +23 enhancer and this was predicted to be pathogenic (Thomas et al., 2021; PMID 34440349). This suggested that there is strong selection pressure to conserve the +23 enhancer in normal hematopoiesis. However, further studies are required to elucidate the dependency on Runx1 enhancers in normal hematopoiesis and leukemia.”

2. On page 5 in the section Mesodermal differentiation... the authors mention the multitude of enhancers in the Runx1 TAD and their changes during differentiation. However, it is interesting to note that in addition to the 33 new ATAC sites during the transition of ESC to HE, as indicated in the MS, 36 other ATAC sites in ESC are lost in HE (Supp. Table 1). It is also worth mentioning that only 4 open chromatin sites (ATAC sites) are unchanged in ESC, HE and HPC (-467, -772, -778 and +128) and 5 ATAC sites are unique to HPC (-42, -48.6, +6.6, +38, +204).

The results section has been modified to highlight the specific interesting changes in chromatin accessibility that the reviewer noticed. Specifically, in reference to the open chromatin sites gained in mesoderm (page 5, line 12-13) “Compared to mESCs, at this mesodermal stage we identified thirty-six open chromatin sites that are lost and thirty-three sites that are gained”. In reference to the five unique sites in HPCs and four sites that are maintained in all three cell types (page 5, line 44-46) “Five peaks are unique to HPCs (-42, -48.6, +6.6, +38, +204) while four peaks are present in all three cell types (-778 [Setd4 promoter], -772 [Setd4 intronic element], -467 [gene desert CTCF site], +128 [Runx1 P2 promoter]).”. The -778 and -772 peaks correspond to the Setd4 promoter and an intronic element within Setd4, which is expressed in all three cell types. The -467 CTCF site is a non tissue-specific CTCF site. The peak located at +128 overlaps the Runx1 P2 promoter which is a CpG island that is accessible in all three cell types.

3. On page 6 line 5 in relation to Figure 3G, the authors mention that those specific E-P interactions (-59, -43, -42, +23, +110) were increased and that interactions with elements extending further in the gene desert were lost. However, it looks as if an interaction at approximately -400 (no coordinates shown in this region) is not lost, but rather increased. I suggest rephrasing the statement to “...interactions with some elements extending in the gene desert were lost.”

We thank the reviewer for pointing out this subtlety in the data. The interactions maintained in the gene desert correspond to interactions with CTCF sites, while the lost interactions correspond to tissue-specific enhancers. The text has been modified (page 6, line 9-10) and now highlights the fact that “interactions with elements extending further in the gene desert were **decreased generally but maintained at non-tissue-specific CTCF sites in the gene desert**”

4. On pages 6-7 the authors determined the effect of deleting the P1 and P2 promoter-proximal CTCF sites in ESC, on promoter-enhancer (E-P) interactions, Runx1 expression, transcriptome changes in other genes, and hematopoietic differentiation. They show that despite perturbed Runx1 chromatin architecture, resulting from the absence of conserved promoter-proximal CTCF sites, there was no change in P1 and P2 alternative usage, specific Runx1 E-P interactions were maintained, Runx1 expression level was only slightly (and not significantly) reduced and hematopoietic differentiation was unaffected. Yet, deletion of the Runx1 P2 proximal CTCF site resulted in a significant change in expression of 168 genes in the mesoderm, suggesting a delay in the in vitro differentiation of ESC to mesoderm and HPC. While the importance of Runx1 gene dosage on its biological function in hematopoiesis in vivo is a well- documented, both in Runx1/+ heterozygous mice and RUNX1 haploinsufficient humans, it is unclear how such a minor insignificant decrease in Runx1 level in Runx1 P2-proximal CTCF-KO can cause a notable gene expression changes in mesodermal cells. It is also unclear whether there is any decrease in Runx1 at the protein level in these cells. The reader will be interested to know whether deletion of Runx1 P2-proximal CTCF in ESC derived from Runx1/+ heterozygous mice (presumably causing a 50% reduction in Runx1 level) will affect mesodermal gene expression during their in vitro differentiation to the same extent or higher, compared to ESC derived from Runx1 WT mice. It would also be of interest to find out whether Runx1 P2-proximal CTCF-KO mice will display hematopoietic defects in the megakaryocytic lineage as occur in Runx1/+ mice. These points should be addressed and discussed.

The reviewer raises several interesting points related to *Runx1* expression levels. Regarding the question how a small change in *Runx1* can lead to notable changes in gene expression in Flk1⁺ cells, this is perhaps less surprising than it seems. Given that the level of *Runx1* mRNA in Flk1⁺ cells is much lower than that seen in hematopoietic cells (Figure 1e), it is plausible that a small decrease brings it below the threshold needed for hematopoietic development to proceed normally. We feel confident that the decrease extends to Runx1 protein, given that known targets of Runx1 are among the decreased genes including *Ikzf1* (Iacovino, M., et al. 2011; PMID: 21170035), *Erg*, and *Flt1* (TRANSFAC database, Matys et al 2006; PMID 16381825)). We have added this consideration to the results on page 7 (line 46-48), where we state that “several hematopoietic markers **and Runx1 target genes** were downregulated (Figure 5f, **and Supp. Figure 13**, adjusted $p < 0.05$) (Harman et al. 2021, Iacovino et al. 2011, Matys et al. 2006).”. As well, we now include additional hematopoietic genes that are downregulated in P2-CTCF-KO mesoderm in Figure 5f (*Erg* and *Flt1*), and have added an additional Supplementary Figure (Supp. Figure 13), which lists thirty-six genes predicted to be RUNX1 target genes in a KMT2A-AFF1 leukemia gene regulatory network (Harman et al 2021; PMID 34088716) that were downregulated in P2-CTCF-KO Flk1⁺ mesoderm.

The second question is whether we expect that deletion of the *Runx1* P2-proximal CTCF site in *Runx1*^{+/-} ESC would have a more pronounced effect on mesodermal gene expression during in vitro differentiation. Based on our analysis of an attenuated P2 mouse model (P2neo) we indeed expect this would more severely affect the normal development of Flk1⁺ cells. In Bee et al., 2010 (PMID 20139099) we found that mouse embryos carrying one (non-functional) Runx1-rd allele and one attenuated P2

allele (effectively only transcribing *Runx1* from the P1 promoter) died around E12.5 showing a phenotype similar to *Runx1* null embryos. While we do not expect the P2 CTCF deletion to reduce *Runx1* expression as much as the P2neo allele, we would expect its phenotype to be more pronounced on a *Runx1*^{+/-} than on a wild type background. We have added this consideration to the discussion on page 10 (line 1-4), suggesting that “A more substantial phenotype may be seen in a *Runx1* heterozygous background, similar to what was observed in mouse embryos carrying an attenuated *Runx1* P2 allele and a non-functional *Runx1* allele, where the compound phenotype was more severe than that of a homozygous attenuated P2 (Bee et al., 2010).”

We thank the reviewer for drawing attention to the interesting question about the impact of altered *Runx1* levels on megakaryocytes. New analysis of our FACS data revealed that the generation of CD41^{hi} megakaryocytes in the ESC differentiation cultures is not obviously impacted by P1/P2-CTCF KO (see updated Supp. Figure 8 and legend).

Reviewer #3 (Remarks to the Author):

Runx1 is a transcription factor that plays a crucial role in definitive haematopoiesis. Genetic events that affect the RUNX1 gene in humans contribute to the disease pathology of myeloid leukaemias. Therefore, control of Runx1 expression is of high interest to clinicians and biomedical scientists researching blood development and leukaemia. In this interesting paper, Owens et al investigate the contribution of 3D genome organisation to Runx1 expression in mouse ES cells, during a process in which these cells are induced to differentiate into mesoderm and then into haematopoietic progenitor cells (HPC). They found that connections from the two Runx1 promoters alter as the cells differentiate, but that deletion of sites that recruit CTCF at each of the two promoters of Runx1 make little difference to enhancer-promoter connections, with the main effect being a delay in haematopoietic differentiation. This is a beautifully conducted study that was a pleasure to read, and my comments are limited to data interpretation and presentation.

Major comments

When illustrating how the connections change between the Runx1 promoters and the enhancers, it would be good to have consistency in the schematics underneath the TAD diagrams in the figures (2-3, under the 'a' panel of each). Each figure should represent the same set of enhancers, rather than a subset. The enhancers shown should exactly reflect those shown in Fig 1a.

We thank the reviewer for pointing this out. Enhancer labelling has been modified to be consistent across all figures. Please see our updated Figure 1a, Figure 1g, Figure 2a, and Figure 3a.

Because the P1 and P2 CTCF motifs are in the same (reverse) orientation, you wouldn't expect CTCF-anchored loops to be responsible for the interactions between P1 and P2 (same-oriented CTCF sites don't stop loop extrusion), ie the sub-TADs aren't likely to be CTCF-dependent anyway. The contacts that show up on the tiling Hi-C could be being mediated by TFs or compartment formation (clustering of like histone modifications).

The discussion has been modified and now includes additional citations to incorporate the reviewers' comment. Specifically, on page 9 (line 21), we now state that “The **fact that P1 and P2-CTCF sites are not in a convergent orientation, and the fact that** sub-TADs/GADs **form** within the gene-body of actively transcribed genes, suggests that instead of being caused by a CTCF-dependent mechanism like loop extrusion, these structures may be dependent on **a combination of factors including** transcriptional processes (Zhang et al. 2020), **TF binding** (Hsieh et al. 2020, Hua et al. 2021), and **clustering of like histone modifications** (Ruthenburg et al. 2007).”

It is worth noting, however, that promoters do also act as chromatin boundaries (e.g. Bozhilov et al. 2021; PMID: 34155213), as mentioned in response to point 2 of reviewer one's comments. Moreover, non-convergent CTCF sites do interact, albeit with a lower frequency (8% of interactions between non-convergent CTCF sites in Rao et al. 2014; PMID 25497547). Therefore, we feel that the non-typical CTCF orientation observed at *Runx1* promoter-proximal CTCF sites on its own did not exclude the possibility of the interaction being CTCF-mediated. However, the chromatin analyses of P1/P2-CTCF-KO cells we performed did provide evidence that the interactions are at least partially CTCF-independent.

KO of CTCF sites. It's actually not surprising that taking out CTCF sites at P1 and P2 didn't affect expression or E-P contacts very much, because the contacts during differentiation are likely to be driven

by TFs. The data on these CTCF site KO's could be interpreted more specifically. For example - deletion of the P2 CTCF site allows from P2 onwards to the 3' end of the gene to be incorporated into larger looped domains that now terminate near *Clic6*. That is, CTCF would likely bind to the next best 3' CTCF site. In both cases this should expand the loops size that P1 and P2 can reside in, and influence the frequency of local contacts. Deletion of the P1 CTCF site allows P1 to make contacts to the 5' that seem to come at the expense of 3' contacts down to P2 (because the P1-P2 sub-TAD is weakened - P1 seems to fall out of this region).

We agree with the reviewer's detailed interpretation of our data and we provide new explanation of this in the discussion (page 9, line 15-20). We now state that "Promoter-proximal CTCF site deletion facilitated new interactions with CTCF sites distal to the deleted CTCF sites, leading to expanded loop domains at *Runx1*. Namely, deletion of P2-CTCF led to increased interactions between upstream regions and the centromeric end of the TAD up to a cluster of CTCF sites close to *Clic6* (Figure 6). Moreover, P1-CTCF-KO led to increased interactions upstream of the P1 promoter at the expense of downstream contacts in the P1-P2 sub-TAD which was weakened."

The authors should take the opportunity include a model diagram/figure, illustrating pictorially what changes happen in the 3D organisation of the *Runx1* gene as cells differentiate. Diagrams could be included to show the expansion of TADs/loops downstream to near *Clic6* upon deletion of the P2 CTCF site. Expansion of these loops into larger loops could weaken E-P interactions that control transcription bursts at P2, thereby leading to the delay in haematopoietic development that the authors observed.

An additional figure has been prepared (Figure 6) that shows a schematic model of chromatin landscape changes during differentiation and after promoter-proximal CTCF site deletion. This is referred to in the Discussion (page 8, line 17 and page 9, line 18).

Specific comments

Tiling Hi-C - works by having a probe to every unique DpnII fragment. There are some repetitive regions in the mouse *Runx1* TAD. Can the frequency of interactions at repetitive regions be confidently mapped?

We thank the reviewer for drawing attention to this relevant technical point. Stringent BLAT filtering is applied during probe design such that only oligos that are highly specific and likely to bind once in the genome are used. As a consequence, repetitive regions will not be captured by oligos during the Tiled-C procedure. However, repetitive sequences might still be ligated to a non-repetitive fragment that was captured and therefore would be sequenced. Any repetitive sequences will then be excluded during read mapping, where only reads that are mapped unambiguously to the genome are retained. As a result of retaining only unambiguously mapping reads, repetitive DNA sequences such as those in the *Runx1* TAD will have no reads mapped to them. It is worth noting that repetitive sequences that are not analyzable by Tiled-C would also not be analyzable by Hi-C. However, as read counts are binned, each 2kb bin typically contains multiple DpnII fragments, and any bin that contains a repetitive region may still have reads unambiguously assigned to it owing to the non-repetitive DpnII fragments that are contained within that bin. Below is a plot showing that the vast majority of the *Runx1* TAD is covered by DpnII fragments that are captured by stringently filtered Tiled-C oligos (Rebuttal Figure 2).

To further clarify these details, the methods section has been modified (page 12, line 23-24) to state that "Probe sequences were stringently BLAT-filtered to exclude repetitive sequences, and synthesized

in-house (Oudelaar et al. 2020).” We now also state that “PCR duplicate-filtered bam files **containing uniquely mapping reads** were converted to sam files (samtools) and then into sparse raw contact matrices” (page 12, line 35-36).

Rebuttal Figure 2. UCSC screenshot showing DpnII fragments captured in Tiled C

Figure 1 d-f - there seem to be different numbers of replicates for undifferentiated, mesoderm, and HPC samples - explanation for this?

ES cell differentiations can be variable and do not always provide sufficient cell numbers for downstream analyses. To compensate for this, we routinely start 4 differentiations if 3 replicates are needed. For robustness we still process all replicates that yield sufficient cells for our analyses. Hence the variation in the number of replicates.

Fig 1e, Flk1 is the term used for this gene in the article text while Kdr is used in the figure, please make consistent between the two.

We thank the reviewer for highlighting this discrepancy in labelling of the Flk1/Kdr gene. Figure 1e has now been updated and reads “Flk1” in line with the text.

Fig 1g, the lack of annotated 'forward' CTCF motifs under P2 suggests that the loop/domain between P2 and the *Clic6* boundary is not CTCF-dependent, at least not at the 5' end. Do the authors have alternative explanations?

This point is well taken. It is true that CTCF sites are typically thought to interact with other CTCF sites, but recent work also shows the tendency for CTCF sites to interact with promoters (Hua et al. 2021; PMID: 34108683). Indeed, this agrees with the previous findings in the literature that TAD boundaries are often associated with promoters and regions of active transcription (Dixon et al. 2012; PMID: 22495300; Rao et al. 2014; 25497547). Further, recent work that was published after we submitted our manuscript demonstrates that active promoters can act as orientation-dependent chromatin boundaries (Bozhilov et al. 2021; PMID: 34155213). Therefore, we believe that the interaction seen between *Runx1* P2 and the *Clic6* boundary CTCF sites may, at least in part, be mediated by the P2 promoter itself. Indeed, our data support this notion, as the P2 promoter region still interacted with the *Clic6* boundary CTCF sites in P2-CTCF-KO cells (Figure 4d, top panel). We added the new citation to the discussion (page 9, line 28-31) and clarified that we found that “as reported for other promoters (Bozhilov et al. 2021, Cho et al. 2018, Harrold et al. 2020, Schwessinger et al. 2020), the *Runx1* promoters may function as chromatin boundaries in a CTCF-independent manner, which would explain the residual sub-TADs **and E-P interactions, as well as the expanded chromatin interactions** observed in P1/P2-CTCF-KO cells”

Fig 2a, enhancers and ATAC peaks, the enhancer annotations are not all the same as those listed in the

red circles in fig 1a. There seems to be extra ones in fig 2a. In fact, it would be good to keep all elements consistent between figs 1-3 in the diagrams beneath the TAD images.

Thanks for picking this up. Enhancer labelling has been modified to be consistent across all relevant figures, which has made the results clearer. Please see updated Figure 1a and 1g, Figure 2a, and Figure 3a.

Fig 2g P1 appears to also interact more with +23 upon differentiation to mesoderm.

The reviewer rightly points out the small increase in interactions between *Runx1* P1 and +23 enhancer when cells differentiated to mesoderm. As such, the results section has been modified (page 5, line 30-32) and now states that “a slight specific increase was seen in interactions between P1 and elements -181 and -171 in the gene desert, and elements +23, +48, +110 within *Runx1* intron 1”.

Page 5: "early spatiotemporal control of *Runx1* expression at the onset of hematopoiesis is associated with increased CTCF interactions" - the authors haven't determined that these interactions are mediated by CTCF. They could be TF-mediated or compartment-mediated.

The original sentence referred to increased interactions between CTCF sites, which were robust (Figure 2D). We have now made this clearer, and the statement now reads "early spatiotemporal control of *Runx1* expression at the onset of hematopoiesis is associated with increased interactions between CTCF sites" (page 5, line 34).

Fig 3a, it would be helpful to label in the diagram under the TAD image, the more distal enhancers, -371, -368 etc. It looks like P2 hangs onto one of these when differentiating to HPCs.

New enhancer labelling has been added so that all Figures 1-3 have the same enhancers labelled, including -371, -368 etc in the gene desert. The P2 does indeed interact in the gene desert—likely in the region with the CTCF sites, which will now be clearer to the readers. A comment has also been added to the text on page 6, line 9-10.

Page 7: "highlighting the tissue-specific nature of this CTCF site." should rather refer to the differential binding of CTCF at this site, because the DNA motif remains unchanged.

This is a point well made. The sentence on page 7 (line 14) has been modified and now states "highlighting the tissue-specific binding of CTCF to this site"

Fig 4a, It would be good to reproduce the schematic from Fig 1g (given below the TAD diagram) showing specifically which of these CTCF sites was deleted.

Figure 4a has now been modified to include the schematic of CTCF binding in the wider *Runx1* TAD from Figure 1g. The legend to Figure 4 (page 24, line 33-35) now reads “a) Schematic of *Runx1* TAD showing CTCF binding in mESCs (Vierstra et al. 2014) and the orientation of CTCF motifs underlying peaks. P1 and P2 promoter-proximal CTCF sites are indicated with CRISPR/Cas9 strategies to delete them.”

Fig 4b, P2-CTCF KO, the enrichment of 'pink' pixels downstream of this deleted CTCF site indicates expansion of interactions such that the *Runx1* gene down to near *Clic6* is now contained within several of the upstream anchored loops.

We thank the reviewer for highlighting this interesting interpretation of our data. The results section has been modified (page 7, line 17-18), and states that “Indeed, loss of P2-CTCF led to the region between *Runx1* and *Clic6* interacting ectopically with upstream regions (Figure 4b, upper panel).”

Fig 4c, P1-CTCF KO, there's a loss of interactions in the P1-P2 intron and a gain of interactions proximal to P1 at the 5' end of the site - this could be due to loss of shielding of P1 from nearby 5' enhancers.

This certainly is a valid interpretation of the data and a plausible suggestion for what could be causing the expanded interactions seen for P1 in P1-CTCF-KO cells. We have modified the relevant section of the discussion (page 9, line 29-31) to now state that “the *Runx1* promoters may function as chromatin boundaries in a CTCF-independent manner, which would explain the residual sub-TADs and E-P interactions, as well as the expanded chromatin interactions observed in P1/P2-CTCF-KO cells.”

Fig 5d, stretching it a bit to say that the CTCF-P2 KO is clustering dramatically differently, because the other conditions don't cluster or separate out that much.

This is a point well made. The original statement has been modified to now state that “all three P2-CTCF-KO samples were located at the far end of the distribution of samples” (page 7, line 42).

Fig 5f, surprising that pluripotency genes are up in both P1 and P2 CTCF-KO lines in undifferentiated cells, why is this?

The reviewer rightly points out this discrepancy in pluripotency gene expression levels. This is likely not a biologically meaningful result, instead probably being due to a technical limitation. Undifferentiated cell samples from wild type (batch one) and P1/P2-CTCF-KO clones (batch two) were prepared at different times, making them not directly comparable. To ensure clarity, undifferentiated cell samples and pluripotency genes have been removed from Figure 5f, which now only shows data where samples were prepared at the same time, facilitating direct comparisons to be made between them.

Page 8: " In addition to these preformed chromatin structures, increasing CTCF-CTCF interactions were observed upon *Runx1* activation that might reflect a higher rate or processivity of loop extrusion in the *Runx1* TAD." Very unlikely to be the mechanism, more likely to be TF-driven, for a recent confirmatory ref see PMID: 32213323. Consider the view that TADs merely provide a framework for specific regulatory interactions, like E-P interactions - see PMC8035076

We thank the reviewer for highlighting these papers that we now cite in our manuscript. These interesting alternative explanation of the potential mechanisms behind the chromatin structures that we observe are now discussed. Specifically, page 8 (line 48-49) now suggests that “increased TF binding could be driving the specific increases in chromatin interactions that were observed over differentiation (Hsieh et al. 2020, Hua et al. 2021).” Additionally, the top of page 9 (line 13-15) now indicates that “residual sub-TAD structures may still have provided a framework within which specific regulatory interactions could take place (Ing-Simmons et al. 2021).”

REVIEWERS' COMMENTS

Reviewer #1 (Remarks to the Author):

The authors have addressed most of my concerns, but I still need to know the differentiation potential of ESC to HPC after depletion of P1 and P2-CTCF.

Reviewer #2 (Remarks to the Author):

In their rebuttal letter and revised manuscript, the authors have addressed all the points I raised in my review letter and have changed the manuscript accordingly to my full satisfaction. I recommend publishing it in Nature Communications.

Reviewer #3 (Remarks to the Author):

The authors have satisfactorily addressed my comments, and the manuscript is much improved. Figure 6 is a great addition, but I am a little confused about the dotted lines representing long range interactions - it seems like these dotted lines should be shown to connect with something rather than hang there (though I do appreciate there will be several connections to represent).

Point-by-point response to reviewers' comments

Reviewer #1 (Remarks to the Author):

The authors have addressed most of my concerns, but I still need to know the differentiation potential of ESC to HPC after depletion of P1 and P2-CTCF.

We are pleased we have been able to address most of the reviewer's comments. As detailed in our previous response (text copied below for ease of referral) we still politely disagree with the reviewer on their one remaining concern. We believe that a P1 and P2-CTCF double deletion is not critical to the main topic of our study, i.e. the dynamic 3D changes at the *Runx1* locus during hematopoietic differentiation, mapped by state-of-the art Tiled-C. In addition, exploring possible CTCF redundancy in a conclusive manner would constitute a significant time commitment due to the many CTCF sites in the *Runx1* locus that would need to be deleted systematically. Indeed, we feel strongly that it would not be in the interest of the wider field to delay making our data available.

Text copied from previous point-by-point response:

"While there is general agreement in the field that there can be redundancy between CTCF sites (addressed in, for example, Hanssen et al. 2017; PMID: 28737770), we decided against deleting both P1 and P2 CTCF sites for the following reasons:

*1) The focus of this study is on the dynamic 3D chromatin changes seen at *Runx1* during hematopoietic differentiation, including changes in E-P interactions. Based on the intriguing observation of sub-TAD formation during differentiation, we extended our study to explore a possible role of promoter proximal CTCF sites in this, as promoter proximal CTCF sites are a previously not well studied class of CTCF sites. While in theory CTCF site redundancy might explain the mild phenotype observed with P1 or P2 CTCF site deletion (discussed on page 9, line 44-48), the P1 and P2 CTCF sites are in the same orientation making it unlikely that they are together responsible for the sub-TAD formation, as also commented by reviewer 3. This information has been added to the discussion on page 9 (line 23-27) to read "The fact that P1 and P2-CTCF sites are not in a convergent orientation, and the fact that sub-TADs/GADs form within the gene-body of actively transcribed genes, suggests that instead of being caused by a CTCF-dependent mechanism like loop extrusion, these structures may be dependent on a combination of factors including transcriptional processes (Zhang et al. 2020), TF binding (Hsieh et al. 2020, Hua et al. 2021), and clustering of like histone modifications (Ruthenburg et al. 2007)."*

*2) To properly address the question of CTCF site redundancy, a systematic perturbation of CTCF sites would need to be undertaken. There are thirty-one CTCF sites in the *Runx1* TAD, several of which could play a role in redundantly regulating *Runx1*. Perturbing these would be a large undertaking, as for each (combination of) CTCF site(s), at least three independent clones would have to be generated and fully validated on multiple levels (as was done in Supp. Figures 5 and 6) to ensure the integrity of the locus. We feel this would be a separate project, outside the scope of our study."*

Reviewer #2 (Remarks to the Author):

In their rebuttal letter and revised manuscript, the authors have addressed all the points I raised in my review letter and have changed the manuscript accordingly to my full satisfaction. I recommend publishing it in Nature Communications.

We thank the reviewer for their positive endorsement.

Reviewer #3 (Remarks to the Author):

The authors have satisfactorily addressed my comments, and the manuscript is much improved. Figure 6 is a great addition, but I am a little confused about the dotted lines representing long range interactions - it seems like these dotted lines should be shown to connect with something rather than hang there (though I do appreciate there will be several connections to represent).

We thank the reviewer for their positive assessment and have adapted Figure 6 as requested.